# PCM1 coordinates centrosome asymmetry with polarized endosome dynamics to regulate daughter cell fate

Xiang Zhao [1,2] ✉, Vincent Mouilleau [1], Yiqi Wang[3], Ahmet Can Solak[2,4], Jason Q. Garcia [1], Xinye Chen[1,5], Xiaoyu Shi [6], Christopher J. Wilkinson [7], Loïc A. Royer [2] ✉, Zhiqiang Dong [1,3] ✉ & Su Guo [1] ✉

Vertebrate radial glia progenitors (RGPs) balance self-renewal and differentiation through asymmetric cell division (ACD), which involves unequal centrosome inheritance. How centrosome asymmetry directs cell fate remains poorly understood. Here, we identify Pericentriolar material 1 (Pcm1) as a key player in this process. In zebrafish embryonic RGPs, Pcm1 is asymmetrically associated with Cep83, a mother centrosome marker. Using in vivo time-lapse imaging and nanoscale-resolution expansion microscopy, we detect Pcm1 on Notch ligand-containing endosomes, where it interacts—either directly or indirectly—with Par-3 and dynein. Loss of *pcm1* disrupts endosome dynamics, increasing neuronal differentiation at the expense of RGP self-renewal. Mechanistically, Pcm1 facilitates the transition from Rab5b to Rab11a and promotes the assembly of Par-3 and dynein macromolecular complexes on recycling endosomes. Furthermore, we find conserved PARD3-PCM1-CEP83-RAB11 associations in human cortical brain organoids. Our findings uncover that Pcm1 links centrosome asymmetry to polarized endosome trafficking, thereby regulating RGP fate decisions.

RGPs are the neural stem cells (NSCs) that undergo orchestrated divisions to produce diverse cell types in the vertebrate central nervous system[1–6]. During active neurogenesis in developing vertebrate embryos, most RGPs undergo asymmetric cell division (ACD), resulting in self-renewing and differentiating daughters with prominent asymmetry of Notch signaling[5,6]. Notch asymmetry between daughter cells is a conserved feature in both invertebrates[7,8] and vertebrates[5,6,9,10]. Studies in both Drosophila[7,8] and zebrafish[11,12] have shown that asymmetric distribution of Notch ligand-containing endosomes in the mother cell contributes to Notch signaling asymmetry between daughter cells. In the zebrafish embryonic forebrain RGPs, which mostly divide along the anteroposterior (A-P) embryonic axis, endosomes containing the Notch ligand Delta D (Dld) are enriched toward the posterior side at anaphase. This process is critically dependent on the polarity regulator Par-3 (also known as PARD3 in humans, and Bazooka in Drosophila) and the dynein motor complex, both of which are localized on the endosomes[12]. How they assemble on the Notch ligand-containing endosomes to drive posterior-directed movement is unclear.

[1]Department of Bioengineering and Therapeutic Sciences, Pharmaceutical Chemistry, Programs in Biological Sciences and Quantitative Biosciences, Institute of Human Genetics, Kavli Institute for Fundamental Neuroscience, Bakar Aging Research Institute, University of California, San Francisco, California, USA. [2]Chan Zuckerberg Biohub, San Francisco, California, USA. [3]College of Biomedicine and Health, College of Life Science and Technology, Huazhong Agricultural University, Wuhan, China. [4]Lawrence Livermore National Laboratory, Livermore, CA, USA. [5]Department of Automation, Tsinghua University, Beijing, China. [6]Department of Biomedical Engineering, Developmental and Cell Biology, Chemistry, University of California, Irvine, Irvine, CA, USA. [7]Centre for Biomedical Sciences, Department of Biological Sciences, Royal Holloway University of London, Egham, Surrey, UK. ✉e-mail: xiang.zhao@czbiohub.org; loic.royer@czbiohub.org; dongz@mail.hzau.edu.cn; su.guo@ucsf.edu

Another organelle that displays asymmetry during RGP division is the centrosome, which undergoes semi-conservative duplication during S-phase, resulting in two centrosomes that differ in age and protein composition[13,14]. Following mitosis, the mother centrosome has a higher activity than the daughter in organizing microtubules and seeds the growth of the primary cilium[15,16]. Asymmetric inheritance of centrosomes has been associated with distinct daughter cell fates in neural progenitors[17–19] and in other cell types[20,21], but how centrosome asymmetry confers daughter cell fate differences at the mechanistic level remains unclear.

In animal cells, the centrosome is composed of a pair of centrioles embedded in a cloud of electron-dense pericentriolar material (PCM)[22–24], of which Pcm1 is a core component involved in maintaining centrosome integrity, anchoring microtubules[25,26], and ciliogenesis[27–29]. The human *PCM1* has been implicated in several forms of cancer and neuropsychiatric disorders[27,28,30,31], but the in vivo dynamics and function of PCM1 in mitotic progenitors are unclear. Here, in embryonic zebrafish forebrain RGPs, we reveal a previously unknown asymmetry of Pcm1 between the two centrosomes and its association with Cep83, a marker of the mother centrosome and component of distal appendages (DAPs) of the mother centrioles[32–34]. Moreover, Pcm1 is localized in the endosomes. Using in vivo time-lapse imaging, high-resolution microscopy, and molecular genetic approaches, we further uncover a critical role of Pcm1 in the asymmetric segregation of recycling Notch ligand-containing endosomes; it does so by facilitating the formation of macromolecular complexes composed of Par-3 and dynein/dynactin on the recycling endosomes. In hiPSC-derived neural progenitors from both 2D neural rosette and 3D brain organoid cultures, we uncover the asymmetry of PCM1 and its co-localization with CEP83. Co-localization of PCM1 on the endosomes with PARD3 and the recycling endosomal protein RAB11 is also observed.

## Results

### Pcm1 localization to centrosomes and endosomes in zebrafish forebrain mitotic RGPs

Given the importance of centrosomes in organizing RGP division[17,34], we studied the role of Pcm1, a conserved component of the pericentriolar satellites[25]. Using a custom-generated antibody (see "Methods"), we first examined the subcellular distribution of Pcm1 in mitotic RGPs.

During active neurogenesis (24-30 h post fertilization, hpf), most forebrain RGPs divide along the anteroposterior (A-P) embryonic axis[12]. At metaphase, Pcm1 was distributed in a cloud-like pattern around the γ-tubulin-positive centrosomes. Asymmetry of the cloud along the A-P axis was often detected, with more Pcm1 around the posterior centrosome. By labeling the mother centrosome with Cep83-EGFP mRNA injection, we found Pcm1 was associated with Cep83 and γ-tubulin, always on the posterior centrosome ($n = 28$). Based on its known role in ciliogenesis, our data reveal that Pcm1 is preferentially associated with the mother (posterior) centrosome that seeds primary cilia growth[35] (Fig. 1a, b, d–f).

Pcm1 distribution in mitotic RGPs exhibited two distinct patterns: tightly centrosome-associated and dispersed in the pericentrosomal region (Fig. 1a–c). This same distribution pattern was also observed in interphase RGPs (Supplementary Fig. 1), demonstrating the dynamic nature of Pcm1 localization throughout the RGP cell cycle.

Quite unexpectedly, Pcm1 immunoreactivity was frequently observed between the separating nuclei (Fig. 1b, e), a region that we referred to as the central zone. The central zone harbors central spindles, where active organelle trafficking and sorting take place. Quantification uncovered variable amounts of Pcm1 in the central zone relative to its total immunoreactivity in the cell (Fig. 1g). These variabilities reflect either the dynamics or the heterogeneity of Pcm1 distribution in the central zone.

The specificity of Pcm1 localization patterns was validated using both *pcm1* morphants (MO) and *pcm1* knock-out (KO) embryos derived from homozygous mutant parents generated via CRISPR genome editing (Fig. 1c, Supplementary Fig. 2a–c). Embryonic development was disrupted in both *pcm1* MO and KO embryos (Supplementary Fig. 2d–f).

Given the prominent localization of Pcm1 in the central zone, we wondered whether it might be associated with Dld endosomes, which are previously shown to converge toward the central zone, followed by their posterior-directed movement[12]. To observe the in vivo dynamics of Pcm1 together with Dld endosomes, we generated GFP-tagged Pcm1 (Pcm1-linker-GFP). Pcm1 is a large protein of over 2000 amino acids with alternatively spliced forms reported in the database (ENSEMBL ENSDARG00000062198). We used the isoform pcm1-201 (ENSEMBL ENSDART00000149026). Both N- and C-terminal tagged versions of Pcm1 were generated and tested. GFP-linker-Pcm1, when expressed in the embryos, caused embryonic lethality, suggesting that N-terminal tagged Pcm1 dominantly interfered with embryonic development. The C-terminal tagged Pcm1, which did not interfere with embryogenesis, was used in our study (Supplementary Fig. 2g). The mRNAs encoding Pcm1-linker-GFP were microinjected into one blastomere of 16-cell stage embryos to achieve sparse labeling. At ~22 hpf, an antibody against the Notch ligand Dld was injected into embryonic brain ventricles to label internalized endogenous Dld as previously described[36] (Fig. 1h). Pcm1 expression level varied; in most RGPs, the Pcm1-GFP reporter formed puncta (of variable sizes depending on expression levels). These puncta were in proximity to Dld endosomes and together they moved toward the posterior side (Fig. 1i, j, Supplementary Videos 1–3). By telophase, most Pcm1-GFP and Dld endosomes were enriched in the posterior daughter (Fig. 1k). Together, these results uncover a previously unknown asymmetric distribution of Pcm1 at one of the two centrosomes (the posterior mother centrosome) in mitosis. Moreover, Pcm1 is also localized in the central zone near Dld endosomes at anaphase and becomes preferentially localized in the posterior daughter after ACD.

### Requirement of *pcm1* for asymmetric distribution of endosomes in mitotic RGPs

In zebrafish, asymmetric segregation of Dld endosomes in mitotic RGPs is dependent on the polarity regulator Par-3 and the dynein motor complex[12]. To understand how Pcm1 might contribute to endosome asymmetry, we labeled the internalized Dld as schematized in Fig. 1g and performed in vivo time-lapse imaging in control and *pcm1*-deficient embryos. A significant reduction of dividing RGPs was observed in both *pcm1* MOs and KOs. Intriguingly, perpendicular divisions were reduced with a corresponding increase of non-perpendicular divisions that are often associated with differentiation (Supplementary Fig. 3a–c). In the *pcm1*-deficient RGPs that were able to divide, internalized Dld fluorescence was reduced (Supplementary Fig. 3d).

Despite a reduction of Dld endosomes in *pcm1*-deficient RGPs, enough were present to characterize their intracellular dynamics. In most WT mitotic RGPs, internalized Dld first converged toward the central zone and then moved in a polarized direction toward the posterior side. However, in *pcm1*-deficient RGPs, asymmetric segregation of Dld endosomes into the posterior daughter was significantly reduced. Co-injection of the *pcm1* mRNA with *pcm1* MO restored asymmetric Dld endosome distribution in the posterior daughter cells (Fig. 2a, b, Supplementary Videos 4–7). Analysis of single-endosome dynamics showed that endosome trajectories were not posterior-bound in *pcm1*-deficient mitotic RGPs (Fig. 2c). Their distribution shifted significantly anteriorly (Fig. 2d). Meanwhile, Dld endosomes in the *pcm1* mRNA-rescued RGPs were more posterior-restricted from the beginning of mitosis (Fig. 2c). Moreover, the average velocity of *pcm1*-deficient endosomes was significantly higher than controls,

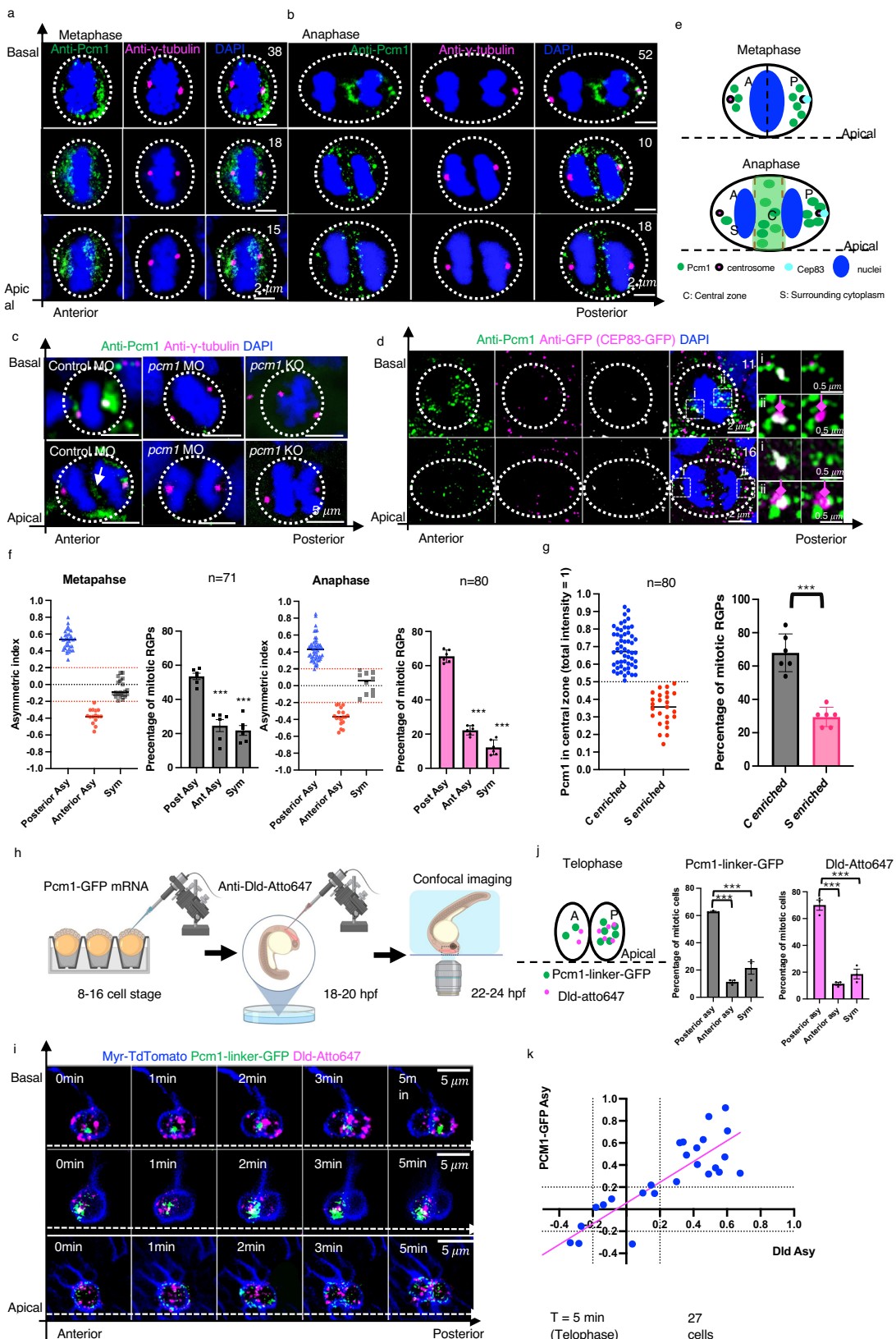

suggesting that the deficiency in posterior-bound endosome movement is due to a lack of directionality, not a deficiency in motility. The *pcm1* mRNA-rescued endosomes restored posterior-bound movement but still showed higher average velocity (Fig. 2e). Together, these results indicate that Pcm1 is required to promote posterior-directed polarized dynamics of Dld endosomes.

Consistent with these observations, we noted that the expression of the Notch signaling effector gene *her4.1* was significantly reduced in the forebrain of the *pcm1* morphant (Supplementary Fig. 4a), and such a deficit was rescued by delivery of the *pcm1* mRNA (Supplementary Fig. 4a, c).

**Fig. 1 | Pcm1 is enriched around the posterior centrosome and in the central zone of zebrafish embryonic forebrain mitotic RGPs. a** Metaphase RGPs with posterior enrichment (upper row, $n = 38$), anterior enrichment (middle row, $n = 18$) and symmetric (lower row, $n = 15$) peri-centrosome localization of Pcm1. **b** Anaphase RGPs with Pcm1 in the central zone (Top: intense labeling, $n = 52$; middle: scattered labeling with posterior enrichment, $n = 10$; bottom, scattered labeling with anterior enrichment, $n = 18$). RGPs were from eight embryos. Scale bars, 2 μm. **c** Mitotic RGPs from control MO, *pcm1* MO, and *pcm1* KO embryos. Three independent repeats with similar results. Scale bars, 5 μm. Maximum intensity projection (MIP) of 8-10 z-plane (0.26 μm z-step). **d** Pcm1, CEP83-EGFP, and γ-tubulin in mitotic RGPs at metaphase ($n = 11$) and anaphase ($n = 16$). The pair of centrosomes of each RGP was outlined, and enlarged views were shown in i & ii. MIP of 10 z-planes are shown, Z-step is 0.26 μm. Scale bars indicate 2 μm or 0.5 μm (i & ii). Magenta arrows indicate the mother centrosome. **e** Schematic of Pcm1 localization in mitotic RGPs. **f** Statistic of Pcm1 pericentriolar asymmetry in metaphase ($n = 71$) and anaphase ($n = 80$) RGPs. **g** Statistics of Pcm1 distribution in the central zone of anaphase RGPs ($n = 80$). In (**f**) and (**g**), *** indicates $p < 0.0001$. Two-tailed unpaired *t*-test, six independent repeats. Error bars indicate SEM. **h** Schematic of in vivo imaging of Pcm1-GFP. Created in BioRender. Zhao, X. (2025) https://BioRender.com/ps32kwp. **i** Three montages show typical Pcm1-GFP dynamics with Dld endosomes. In all images, MIP of 5 μm z-stacks (1 μm z-step) are shown. Six independent repeats with similar results. **j** Statistics of Pcm1-GFP and Dld A-P asymmetry at $T = 5$ min (telophase). $N = 27$ from 6 embryos. *** indicates $p < 0.0001$. Two-tailed unpaired *t*-test. Error bars indicate SEM. **k** Scatterplot of individual RGP's asymmetry indices for internalized Dld endosomes (x axis) and Pcm1-GFP (y axis), the dotted lines indicate the asymmetric index threshold: <−0.2: anterior enriched; >0.2: posterior enriched. $N = 27$ from 6 embryos. Magenta line is a simple linear regression test, F = 86.96; R-squared = 0.7767, Sy.x = 0.1887.

## Requirement of *pcm1* for regulating progenitor division modes and daughter cell fate revealed at clonal resolution

To determine the impact of Pcm1 on RGP division modes and daughter cell fate, we used a transgenic line *Tg[HuC-GFP;Ef1a-H2B-mRFP]*, which marks neurons green and all nuclei red (Fig. 3a). Quantification of neuron numbers relative to total nuclei in the telencephalon showed a significant relative increase of neurons in *pcm1*-deficient embryos (Fig. 3b), although the overall number of nuclei appeared decreased (Fig. 3a), consistent with reduced proliferation observed in *pcm1*-deficient embryos.

To determine whether the relative increase of neuronal production in *pcm1*-deficient embryos is due to defects in the asymmetry of Dld endosomes resulting in altered modes of cell division, we performed clonal analysis of RGP division. One-cell stage *Tg[HuC-GFP]* embryos were injected with either control or *pcm1* MO, followed by H2B-mRFP mRNA injection into one blastomere at the 16-32-cell stage. In vivo long-term time-lapse imaging was carried out to track mitotic RGPs for ~10 h (Fig. 3c). Three types of divisions were observed (Fig. 3d–e, Supplementary Videos 8–10): 1) progenitor/progenitor (P/P), where both daughters remained HuC-GFP⁻ 10 h after division. Such division could be either symmetric or asymmetric, generating two progenitor daughters that have equal or unequal proliferative potential, as we have previously shown[6]. 2) progenitor/neuron (P/N), where one daughter remained a progenitor and the other became a neuron. Such a division is expected to be asymmetric. 3) neuron/neuron (N/N), where both daughters became HuC-GFP⁺. Such division could also be either symmetric or asymmetric, depending on whether the same or different types of neurons are generated. In control MO-injected embryos, 42% of RGP divisions were P/P, 39% P/N, and 19% N/N ($n = 72$). In *pcm1* MO embryos, there was a decrease of P/P and P/N divisions and a corresponding increase of N/N divisions, which now accounted for 49% of observed divisions ($n = 45$) (Fig. 3d–e). We also found a significant decrease in *her4.1* expression in the forebrain of the *pcm1* MO embryos compared to controls (Supplementary Fig. 4).

Additionally, we performed EdU pulse-chase experiments in *pcm1* KO embryos (Supplementary Fig. 5a). Consistent with a reduced number of mitotic RGPs (Supplementary Fig. 2a), we found decreased EdU labeling, indicative of fewer cells entering S-phase in *pcm1*-deficient embryos (Supplementary Fig. 5b, d). After a 24-h chase, the proportion of EdU⁺Hu⁺ in all EdU⁺ cells was significantly increased in both the forebrain and spinal cord, suggesting an increased neuronal production at the expense of progenitors in *pcm1*-deficient embryos (Supplementary Fig. 5c, e). Overproduction of neurons at the expense of progenitors has also been observed in *pcm1*-deficient mice[37]. Together, findings in zebrafish reveal at an unprecedented clonal resolution that Pcm1 maintains RGP progenitor fate by promoting P/P and P/N division and/or suppressing the N/N division. This is likely due to Pcm1's role in promoting the asymmetric distribution of Dld endosomes (as shown in Fig. 2): In the absence of Pcm1, Notch signaling is defective, thereby resulting in a decrease of P/P and an increase of N/N divisions.

## Co-localization of Pcm1 with Par-3, Rab5b, and Rab11a on Dld endosomes revealed via high resolution expansion microscopy

To understand how Pcm1 might promote posterior-directed Dld endosome movements, we analyzed its sub-cellular co-localization with other proteins in mitotic RGPs, first by conventional immuno-fluorescent microscopy (Supplementary Fig. 7), then high-resolution expansion microscopy (Figs. 4, 5). In developing zebrafish forebrain mitotic RGPs at anaphase, Pcm1 was detected in the central zone, where the polarity regulator Par-3, Dld endosomes, the dynein complex component Dlic1, the dynactin subunit P150, and the recycling endosome marker Rab11a and the early endosome marker Rab5b were also detected. The presence of Rab5b in the central zone was much less compared to Rab11a, suggesting that the central zone mostly harbors recycling endosomes (Supplementary Fig. 7a, b). Quantification of the co-localization coefficient[38] among pairs of proteins uncovered co-localization of Pcm1 with many of these proteins in both the central zone and the surrounding cytoplasm (Supplementary Fig. 7c). Both Pcm1 and Par-3 showed more co-localization with Rab11a in the central zone and with Rab5b in the surrounding cytoplasm. In vivo co-immunoprecipitation further showed that Pcm1 interacted with Par-3, Dlic1, and Dld, Rab11a, and Rab5b, either directly or indirectly (Supplementary Fig. 8).

To visualize protein co-localization at a higher resolution, we performed label retention expansion microscopy (LR-ExM). This technology enables us to detect cytoplasmic Par-3 that decorates Dld endosomes[12]. Endosomes were visualized in projected images of five z-planes (Fig. 4a, right). Intriguingly, Pcm1 intercalated with Par-3 on Dld endosomes (Fig. 4a, Supplementary Fig. 7a). Dld endosomes decorated with both Par-3 and Pcm1 were significantly more in the central zone than in the anterior surrounding cyto-plasm. There were significantly more Dld endosomes in the central zone than in the anterior or posterior surrounding cytoplasmic areas in the RGPs at anaphase (Fig. 4c). The composition of Dld endosomes in the anterior surrounding cytoplasm was significantly different from that of the central zone, with fewer Pcm1-Par3-Dld-positive endosomes, while no significant differences were observed between the central zone and the posterior surrounding cytoplasm (Fig. 4d). These results uncover that most Dld endo-somes in the central zone and posterior surrounding cytoplasm have both Pcm1 and Par-3.

Using Rab5b and Rab11a to visualize the early and recycling Dld endosomes, respectively, we detected an association of Pcm1 with both markers on Dld endosomes (Fig. 5a, d, Supplementary Fig. 9 for more examples and additional markers, including P150, a subunit of the dynactin complex). We found that Rab5b⁺ endo-somes showed a similar distribution ratio among different areas in the anaphase RGPs (Fig. 5c) while Rab11a+ endosomes showed a significantly higher distribution ratio in the central zone compared to the anterior and posterior surrounding cytoplasmic areas (Fig. 5e). Quantifications uncovered that Pcm1⁺Rab5b⁺ Dld

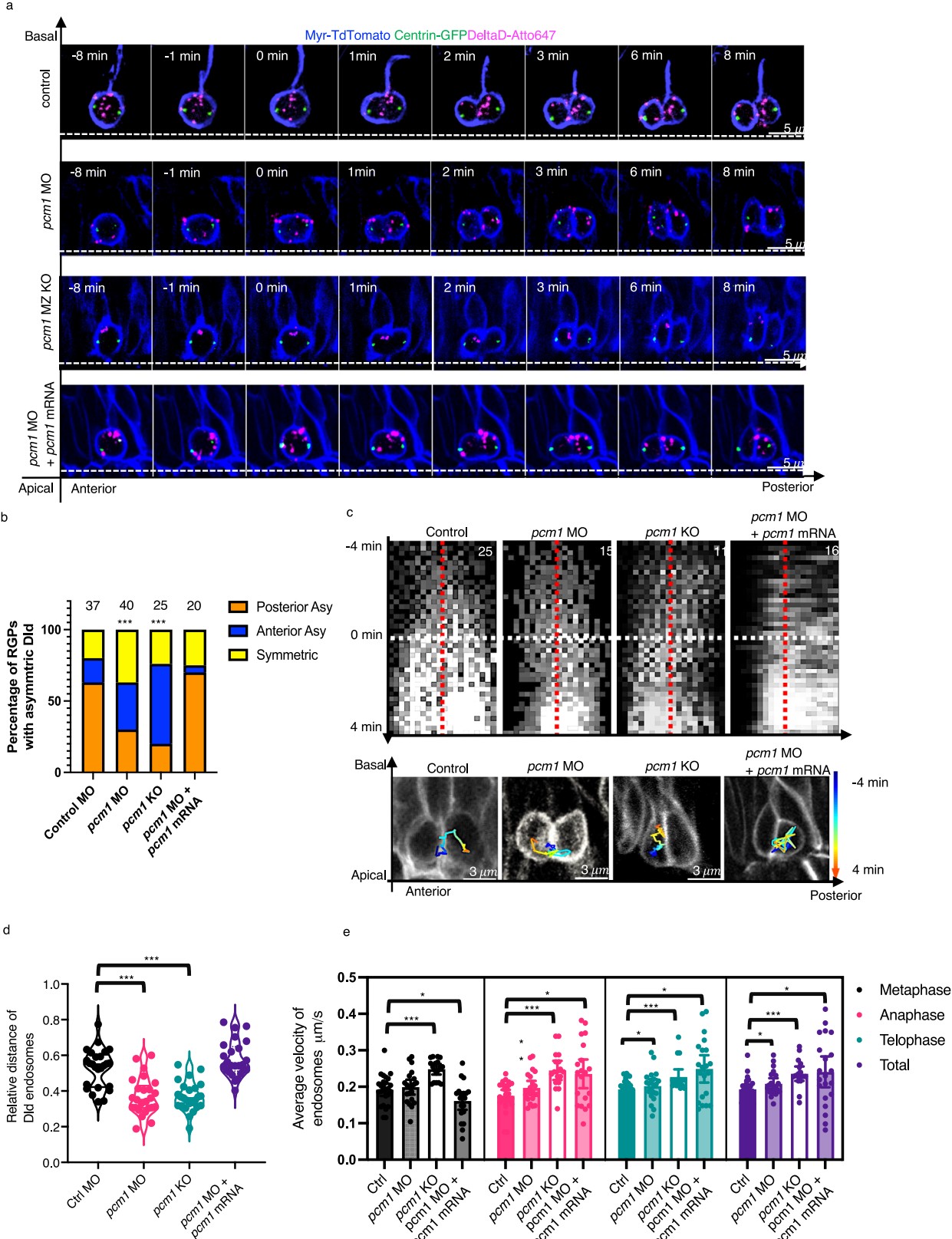

endosomes were significantly depleted in the central zone compared to the anterior surrounding cytoplasm (Fig. 5c), whereas Pcm1+Rab11a+ Dld endosomes were enriched in the central zone compared to the anterior surrounding cytoplasm (Fig. 5f). These results reveal a relative enrichment of Pcm1+ Dld recycling endosomes in the central zone.

## Requirement of Pcm1 for the formation of Par-3-Dynein macromolecular complexes on Dld endosomes in the central zone

To determine what role Pcm1 might play on Dld endosomes, we performed high-resolution expansion microscopy and analyzed anaphase/telophase RGPs in *pcm1*-deficient embryos. In control, most Dld endosomes were decorated with both Par-3 and the dynein complex

**Fig. 2 | In vivo time-lapse imaging reveals the requirement of Pcm1 for posterior-directed Dld endosome movement in zebrafish embryonic forebrain mitotic RGPs. a** Time-lapse imaging montages in control MO, *pcm1* MO, *pcm1* KO and *pcm1* MO + *pcm1* mRNA. The membrane is marked with Myr-TdTomato (pseudo-colored blue), centrosomes are marked with Centrin-GFP (pseudo-colored green), and internalized Dld endosomes are marked with Dld-atto647 (pseudo-colored magenta). T = 2 min denotes the onset of telophase (determined by the membrane abscission between the daughter cells). Each fame is the MIP of 5 confocal z-stacks (1 μm z-step). White dashed lines indicate the apical-basal and anterior-posterior axes. **b** Statistics of the distribution of internalized Dld in telophase RGPs. *** indicates $p < 0.0001$, $\chi^2$ test (chi-square = 73.46, df = 4) was applied between control group and other groups. **c** Kymographs (top) and representative trajectories (bottom) of Dld endosomes, tracked from -4 min to 4 min (the starting

point of anaphase is 0 min, indicated by a white dashed line). The red dashed lines on the kymograph images indicate the center of the cell registered by centrosomes. **d** Averaged distribution of Dld endosomes along the anterior (0) and posterior (1) axis; each dot representing the value from one RGP at telophase (T = 4 min). The positions of centrosomes are used for cell registration. *** indicates $p < 0.0001$, $n = 25$ telophase RGPs per group, two-tailed unpaired *t*-test. Error bars indicate SEM. **e** Average velocity of Dld endosomes in dividing RGPs (T = −4 min to T = 4 min). Significantly higher velocity was detected in *pcm1* MO and KO RGPs compared to the control. The T frame interval is 12 s. Each dot represents the average velocity of all Dld endosomes in a single RGP. *** indicates $p < 0.0001$; ** $p < 0.01$; * $p < 0.05$; two-tailed unpaired *t*-test, Control group, $n = 25$ RGPs; *pcm1* MO, $n = 21$ RGPs; pcm1 KO, $n = 17$ RGPs. *pcm1* MO + *pcm1* mRNA, $n = 20$ RGPs. Error bars indicate SEM.

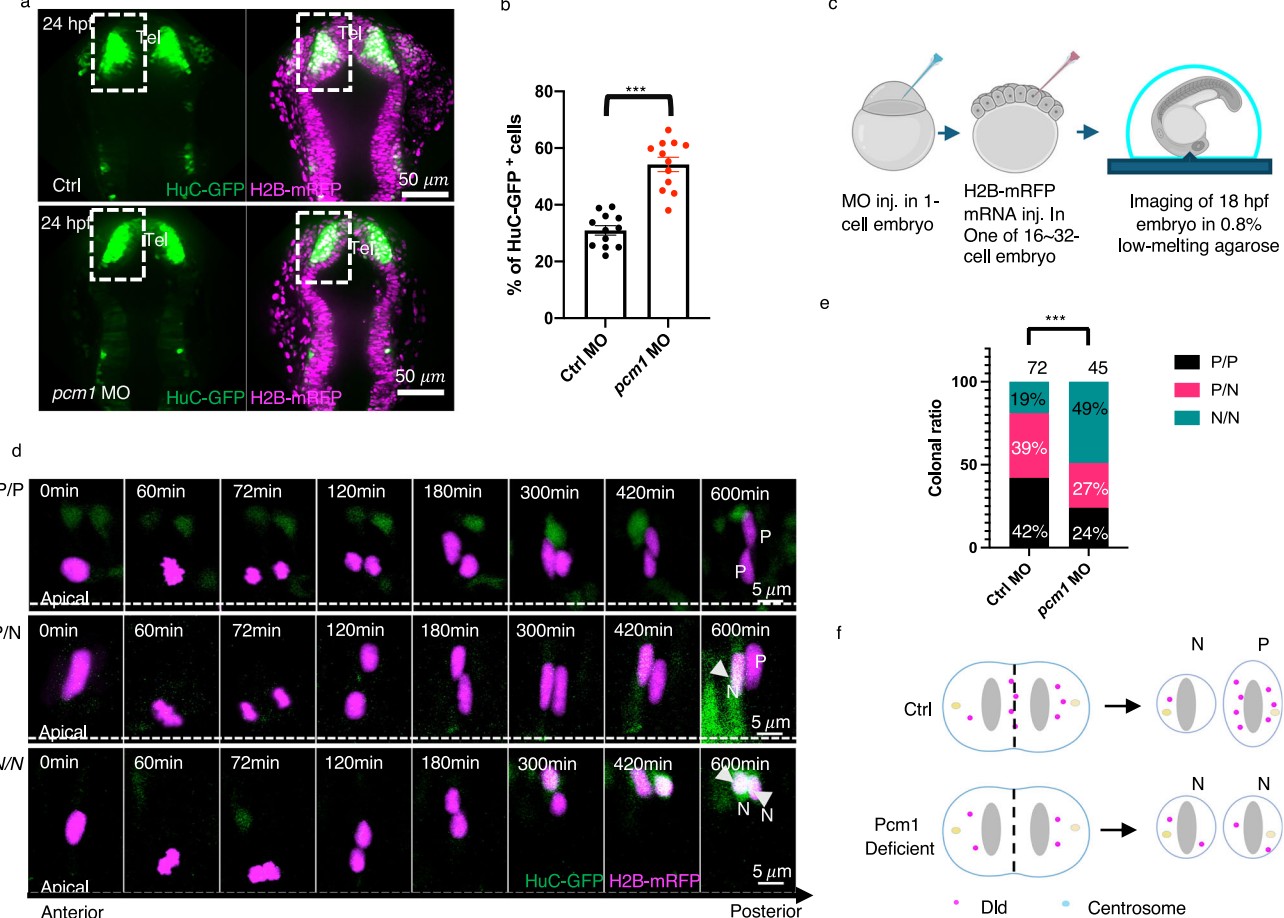

**Fig. 3 | Pcm1 is required to maintain proliferative RGPs in the developing zebrafish forebrain. a** Live imaging of HuC-GFP+ cells of 24 hpf *Tg[HuC-GFP: H2B-mRFP]* embryos. Differentiating neurons in the telencephalon (Tel) are boxed in both control MO and *pcm1* MO embryos. Each image is the MIP of 10 z-planes. The z-step is 1 μm. Scale bars: 50 μm. **b** Statistics show a significant increase of HuC-GFP+ cells relative to the total telencephalic nuclei (H2B-mRFP+) in the boxed area of (**a**). $n = 12$, three embryos from four independent microinjection experiments, *** indicates $p < 0.0001$, two-tailed unpaired *t*-test. Error bars indicate SEM. **c** Schematic of sparse labeling and time-lapse imaging. Created in BioRender. Zhao,

X. (2025) https://BioRender.com/ps32kwp. **d** Montages of time-lapse sequence of images showing clonally labeled RGPs giving rise to progenitor (P) daughter and/or neuron-like (N) daughter cells. Each image is the MIP of 10 z-planes. The z-step is 1 μm. The time interval between each volume of z-stacks is 6 min and the total acquisition time is ~10 h. **e** Statistics show that RGPs in *pcm1* MO embryos underwent more N/N and fewer P/P and P/N divisions. *** indicates p < 0.0001, $\chi^2$ test (Chi-square = 19.46, df = 2). **f** schematic summarizing both the Dld endosome (Fig. 2) and cell fate phenotypes (this Figure).

component Dlic1. In *pcm1*-deficient embryos, however, we observed a significant decrease of Par-3+Dlic1+ Dld endosomes and a corresponding increase of Par-3+-only and Dlic1+-only Dld endosomes (Fig. 6a, b). Moreover, the amount of Dld in these Par-3+-only and Dlic1+-only Dld endosomes was significantly less than that in Par-3+Dlic1+ Dld endosomes, as shown by both staining intensity and surface area (Fig. 6c). These findings suggest that Pcm1 is required to bring together both

Par-3 and Dlic1 on Dld endosomes; in its absence, most Dld endosomes have either Par-3 or Dlic1 but not both. The amount of Dld is further reduced in these Pcm1-deficient Par-3-only or Dlic1-only endosomes. These findings suggest that Pcm1 is required for macromolecular complex formation on Dld endosomes and for the integrity of these endosomes.

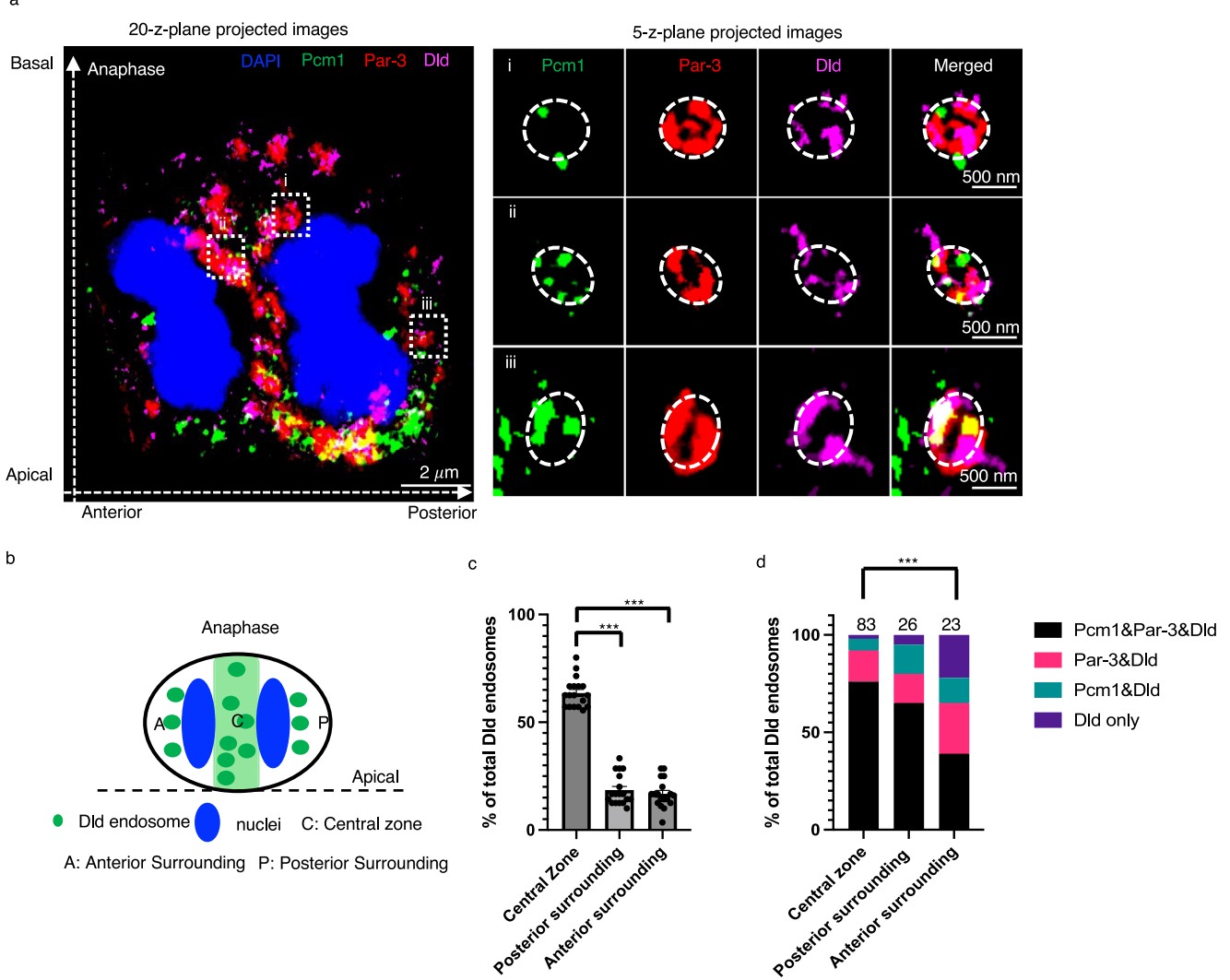

**Fig. 4 | Label retention expansion microscopy (LR-ExM) uncovers Pcm1 colocalization with Par-3 on Dld endosomes in zebrafish embryonic forebrain mitotic RGPs. a** Anaphase RGP immuno-stained with anti-Par-3, anti-Pcm1, anti-Dld and DAPI from 28 hpf embryo cryosections. The left panel shows the whole cell image (MIP of 20 z-planes), and the right panel shows enlarged views (i, ii and iii) of endosomes (MIP of 5 z-planes). The z-step is 0.26 μm. Scale bars denote the biological size. There are six independent repeats of the experiment with similar results. **b** A schematic shows the central zone and surrounding cytoplasm (separated into the anterior vs. posterior) in mitotic RGPs. **c** Statistic of the percentage of DLD-containing endosomes in each RGPs. *** indicates $p < 0.0001$, two-tailed unpaired t-test, $n = 18$, three RGPs from each embryo, six different embryos were used in total. Error bars indicate SEM. **d** Statistics of the percentage of Par-3, Pcm1, and Dld-containing endosomes. All endosomes (83 at the central zone, 23 from the anterior surrounding, and 26 from the posterior surrounding cytoplasm) are from 18 anaphase RGPs. *** indiactes $p < 0.001$, χ² test showed significant difference between the central zone and the anterior surrounding (Chi-square = 33.53, df = 3).

## Requirement of Pcm1 for the trafficking from early to recycling endosomes

Given Pcm1's association with general endosomal proteins, including Rab11a and Rab5b, we reasoned that its role on endosomes might not be limited to those containing Dld. To test this, we analyzed endosomes marked with Rab11a or Rab5b in anaphase/telophase mitotic RGPs. In control RGPs triple-immunostained with Par-3, Rab11a, and P150 antibodies, most endosomes in the central zone had Par-3, Rab11a, and P150. However, in *pcm1*-deficient embryos, Par-3⁺Rab11⁺P150⁺ endosomes were significantly reduced, and a corresponding increase of Rab11a-P150 and Par-3-P150 co-labeling was observed (Fig. 7a, b). These results suggest that Pcm1 is required to bring together Par-3 and P150 on recycling endosomes.

Rab5⁺ endosomes colocalizing with Par-3 and P150 were infrequently observed in the central zone in control anaphase/telophase mitotic RGPs. However, the number of Rab5⁺Par-3⁺P150⁺ endosomes significantly increased in the central zone of *pcm1*-deficient anaphase/telophase mitotic RGPs (Fig. 7c, d). These data suggest that Pcm1 is

required for the trafficking from Rab5⁺ early endosomes to Rab11⁺ recycling endosomes; this deficit possibly contributes to the decreased Rab11⁺Par-3⁺P150⁺ endosomes in the central zone of *pcm1*-deficient mitotic RGPs.

## Dysregulation of genes involved in endocytosis and cell differentiation upon Pcm1 disruption

Consistent with the localization of Pcm1 to endosomes and its function in regulating endosome dynamics, we detected changes in gene expression related to endocytosis and cellular differentiation at the transcriptomic levels upon Pcm1 disruption. RNA-seq was performed on control and *pcm1* KO embryos at ~ 24 hpf. Among the 1176 genes with decreased and 1034 genes with increased expression in the *pcm1* KO embryos, gene ontology (GO) analysis uncovered a significant enrichment for the genes involved in endocytosis and vesicle-mediated transport (Supplementary Fig. 10a). They include genes encoding proteins involved in clathrin exocytosis (e.g., *cplx2l, stxbp5l, anaxa1d, syt7b*), endosomal sorting (e.g., snx5, snx2, *ston2, ap1m2*), or

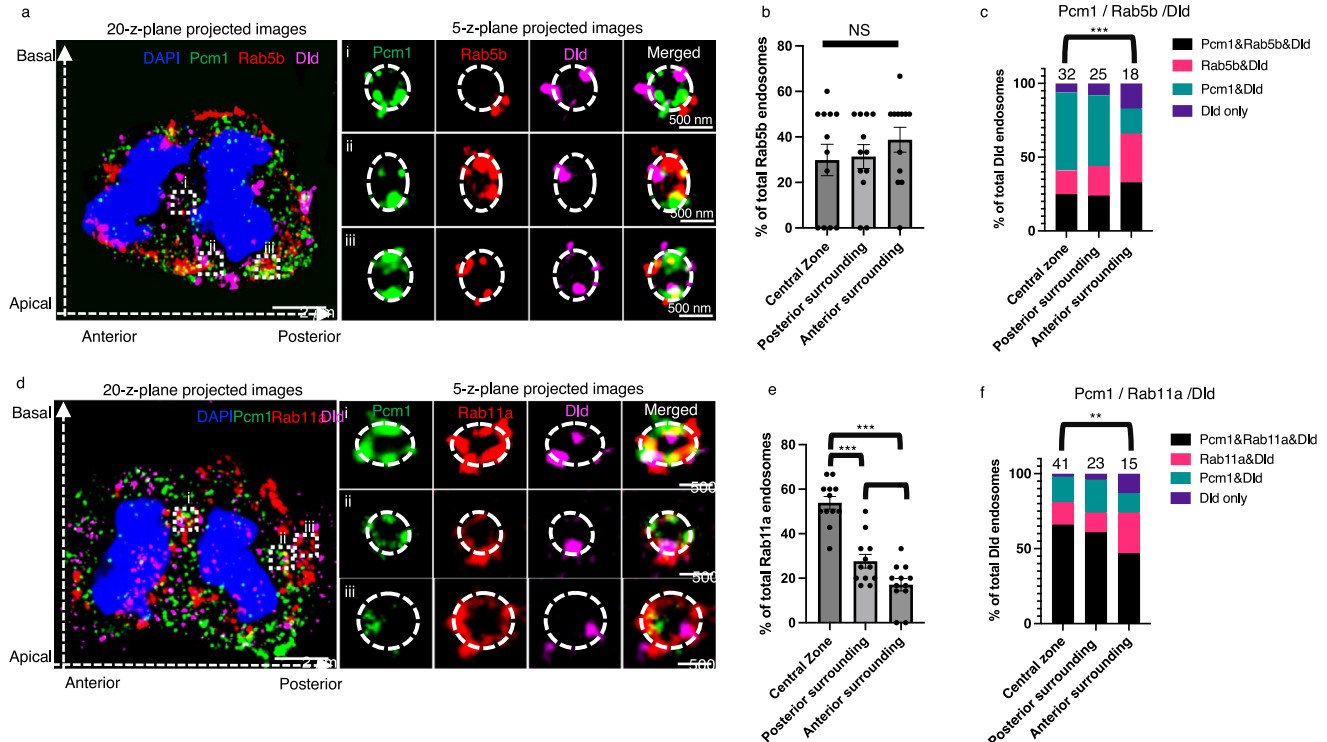

**Fig. 5 | LR-ExM uncovers Pcm1 colocalization with Rab5b and Rab11a on Dld endosomes in zebrafish embryonic forebrain mitotic RGPs.** Anaphase RGP immuno-stained with anti-Pcm1, anti-Rab5b (**a**) or anti-Rab11a (**d**), anti-Dld, and DAPI from 28 hpf embryonic forebrain cryosections. The left panels show a whole cell image (MIP of 20 z-planes); the right panels show enlarged views of endosomes (i, ii, and iii) (MIP of 5 z-planes). The z-step is 0.26 μm. Scale bars denote the biological size. The experiment was repeated four times independently with similar results. **b** Statistics of the percentage of Rab5b+ Dld endosomes identified in each RGP. $n = 12$. Expanded samples from four different embryos were used, and three RGPs from each embryo were chosen randomly for the statistics. One-way ANOVA analyses of three groups showed no significant difference (NS), $p = 0.475$ (Brown-Forsythe test). Error bars indicate SEM. **c** Statistics of the percentage of Rab5b,

Pcm1, and Dld-containing endosomes identified in anaphase RGPs. All endosomes (32 at the central zone, 18 from the anterior surrounding, and 25 from the posterior surrounding cytoplasm) are from 12 RGPs. *** indicates $p < 0.0001$, $\chi^2$ test of central zone and anterior surrounding (Chi-square = 30.78, df = 3). **e** statistics of the percentage of Rab11a+ Dld endosomes identified in each RGP. $n = 12$. Expanded samples from four different embryos were used, and three RGPs from each embryo were chosen randomly for the statistic. * indicates $p = 0.0182$,; *** indicates $p < 0.0001$, t-test was applied between groups, two-tailed unpaired t-test. Error bars indicate SEM. **f** Statistics of the percentage of Rab11a, Pcm1, and Dld-containing endosomes identified in anaphase RGPs. All endosomes (41 at the central zone, 15 from the anterior surrounding and 23 from the posterior surrounding cytoplasm) are from 15 RGPs. ** indicates $p = 0.0016$, $\chi^2$ test (Chi-square = 15.22, df = 3).

localized to endosomes (e.g., *stard3, rab5b, vps8*) (Supplementary Fig. 10b, c).

Together, these results support a role of Pcm1 in regulating endocytosis and endosome dynamics, which is critical for proper progenitor fate and cellular differentiation.

### Asymmetry and colocalization of PCM1 with PARD3 and the recycling endosome component RAB11A in hiPSC-derived neural progenitors

To determine whether the association of PCM1 with endosomes might be a conserved phenomenon, we examined the distribution of PCM1 in hiPSC-derived neural progenitors. Two well-characterized hiPSC lines, WTC11[39] and KOLF2.1J[40], were used for differentiation toward

2D neural rosettes and 3D organoids with dorsal forebrain (cortical) characteristics, adapted from a previously published method[41]. The 3D forebrain organoids were obtained after 25-30 days in culture, and the 2D neural rosettes were harvested for analysis after 21 days in culture (Fig. 8a). We analyzed over 20 neural rosettes and 8 forebrain organoids from each line. This was repeated in four independent differentiation experiments for each line.

Days 25–30, cortical organoids are in a dynamic and critical phase of neurogenesis, including the emergence of neurons and complex neuroepithelial structures. At this stage, the organoids transition from primarily containing neural progenitor cells to a more diverse and organized population of cells[42,43]. Ventricular zone (VZ) RGPs actively

undergo ACD to both self-renew and to produce either a neuron or an intermediate progenitor (IP) marked by TBR2. Importantly, our primary analysis focused on ACD of RGPs along the ventricular lumen surface, rather than quantifying the division of TBR2+ intermediate progenitor populations. The ventricular RGPs are likely comparable to the RGPs we study in the developing zebrafish forebrain.

The cortical organoids consisted of multiple pseudo-stratified ventricular zone (VZ)-like progenitor regions and expressed dorsal telencephalic progenitor markers PAX6, the neural stem/progenitor marker NESTIN, intermediate progenitor cell (IPC) marker TBR2, and nascent neuronal marker HUC (Supplementary Fig. 11a). The neural rosettes expressed PAX6, NESTIN and HUC (Supplementary Fig. 11b). We also validated our custom chicken anti-PCM1 antibody in neural rosettes (Supplementary Fig. 11c), which showed identical staining patterns as a previously published rabbit anti-PCM1 antibody[26].

In cryo-sectioned organoids, PCM1, PARD3, and RAB11A were enriched and co-localized in the VZ-like progenitor regions, whereas only a partial overlap was observed for PCM1, RAB5B, and γ-tubulin (a centrosome marker) (Fig. 8b). A close examination of mitotic neural progenitors at late anaphase or telophase found that most displayed asymmetric distribution of PARD3, PCM1, and RAB11A, but not RAB5B (Fig. 8c, e). Similar to previous findings[44,45], we also noted an increased percentage of progenitors that had their cleavage planes parallel to the VZ surface (Fig. 8d), compared to divisions of zebrafish and mouse embryonic neural progenitors with cleavage planes mostly

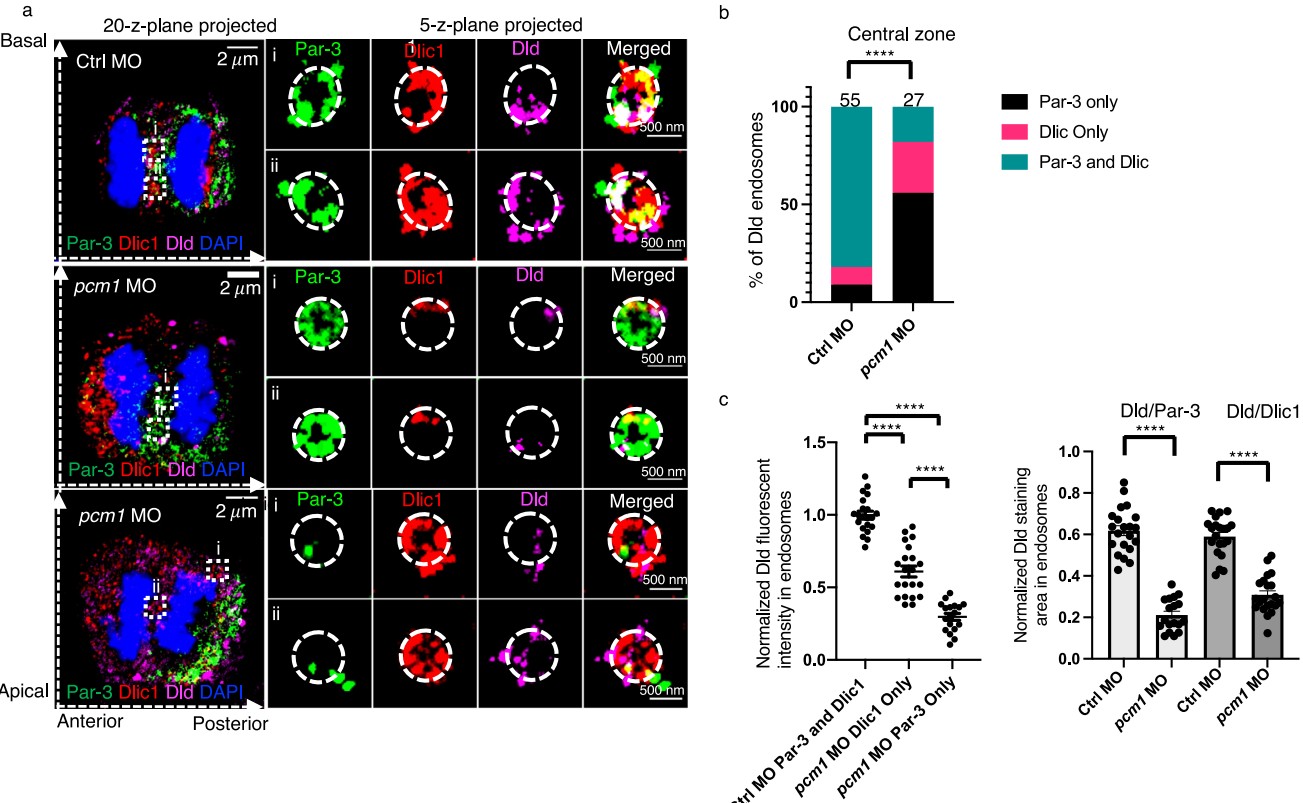

**Fig. 6 | Pcm1 is required for the colocalization of Par-3 and Dlic1 on Dld endosomes in zebrafish embryonic forebrain mitotic RGPs. a** LR-ExM shows Par-3 and Dlic1 on Dld endosomes. RGPs from control MO (upper panel) and *pcm1* MO (lower two panels) are shown. The left panels show whole cell images (MIP of 20 z-planes); the right panels show enlarged views (i and ii) (MIP of 5 z-planes). The z-step is 0.26 μm. The dashed lines with arrows denote anterior-posterior axis and apical-basal axis. Scale bars denote the biological size. **b** Statistics of the percentage of Dld endosomes with Par-3, Dlic1, or both. Endosomes were analyzed from 6 RGPs

of three independent experiments for each group. **** indicates $p < 0.0001$, $\chi^2$ test (Chi-square = 83.20, df = 2). **c** Quantification of Dld fluorescent labeling on endosomes. The anti-Dld immunofluorescent intensity was normalized to that of DAPI (left). Alternatively, Dld staining area on the endosomal surface was normalized with that of either Par-3 or Dlic1 staining areas (right). All endosomes are from six RGPs for each group, collected from three independent experiments. Two-tailed unpaired *t*-test, **** indicates $p < 0.0001$, $n = 21$ endosomes for Ctrl MO and *pcm1* MO Par-3 only group, $n = 18$ for *pcm1* MO Dlic1 only group. Error bars indicate SEM.

perpendicular to the VZ. Quantification of protein co-localization showed the co-localization of PCM1 with PARD3 and RAB11A, but much less with RAB5B, in both the central zone and the whole cell (Fig. 8f). Similar observations were made in the hiPSC-derived neural rosettes (Supplementary Fig. 12). During anaphase, most NPCs (in both 2D rosettes and 3 d organoids) display asymmetric distribution of PCM1, PARD3, RAB11A, but not RAB5B (Supplementary Fig. 13). In interphase NPCs, like in zebrafish, PCM1 displayed pericentriolar enrichment or tight association with centrosomes (Supplementary Fig. 14). Together, these data suggest that the association of PCM1 with PARD3 and the recycling endosome protein RAB11A and their asymmetric distribution in mitotic neural progenitors are conserved from fish to humans.

### PCM1 asymmetry and colocalization with CEP83 on the mother centrosome in mitotic neural progenitors of human cortical organoids

To determine whether PCM1 displays asymmetry on the centrosomes in human neural progenitors, we performed immunofluorescent labeling for PCM1 and CEP83 on KOFL2.1J hiPSC-derived cortical organoids (Day 25 in culture). CEP83 (centrosome protein 83, originally known as CCDC41) is a core component of the mother centriole's distal appendages. CEP83 has been identified as a marker of the mother centrosome and plays an essential role in the formation of the primary cilium by organizing the mother centriole's distal appendages and enabling membrane docking[32,46]. In both metaphase and anaphase mitotic neural progenitors, we detected a significant percentage of

progenitors that exhibited centrosome asymmetry (Fig. 9). Furthermore, in the cases of asymmetric centrosome PCM1, the centrosome with enriched PCM1 was always CEP83⁺ ($n = 117$ mitotic neural progenitors), suggesting that PCM1 is localized on the mother centrosome. Intriguingly, a close examination uncovered that PCM1 and CEP83 were concentrated on distinct centrioles of the mother centrosome.

## Discussion

In this study, we uncovered an asymmetric distribution of Pcm1 at the posterior mother centrosome and its critical role in directing posterior-oriented endosome dynamics during ACD in embryonic zebrafish forebrain RGPs. During metaphase and anaphase, Pcm1 forms an amorphous pericentriolar cloud, asymmetrically distributed around the centrosomes. Notably, Pcm1 is consistently enriched at the posterior mother centrosome, which expresses Cep83, a key marker of the mother centrosome. At anaphase, Pcm1 associates with the Dld-containing endosomes and is essential for their preferential distribution to the posterior, self-renewing daughter cell. Pcm1 facilitates the assembly of macromolecular complexes composed of Par-3 and dynein on Dld endosomes, highlighting its role in polarized endosome dynamics and the maintenance of progenitor fate. Furthermore, we show that PCM1 exhibits centrosomal asymmetry in both zebrafish RGPs and human neural progenitor cells, consistently localizing near CEP83. Regarding endosome dynamics, PCM1 co-localizes with PARD3 and the recycling endosome marker RAB11A. This asymmetric

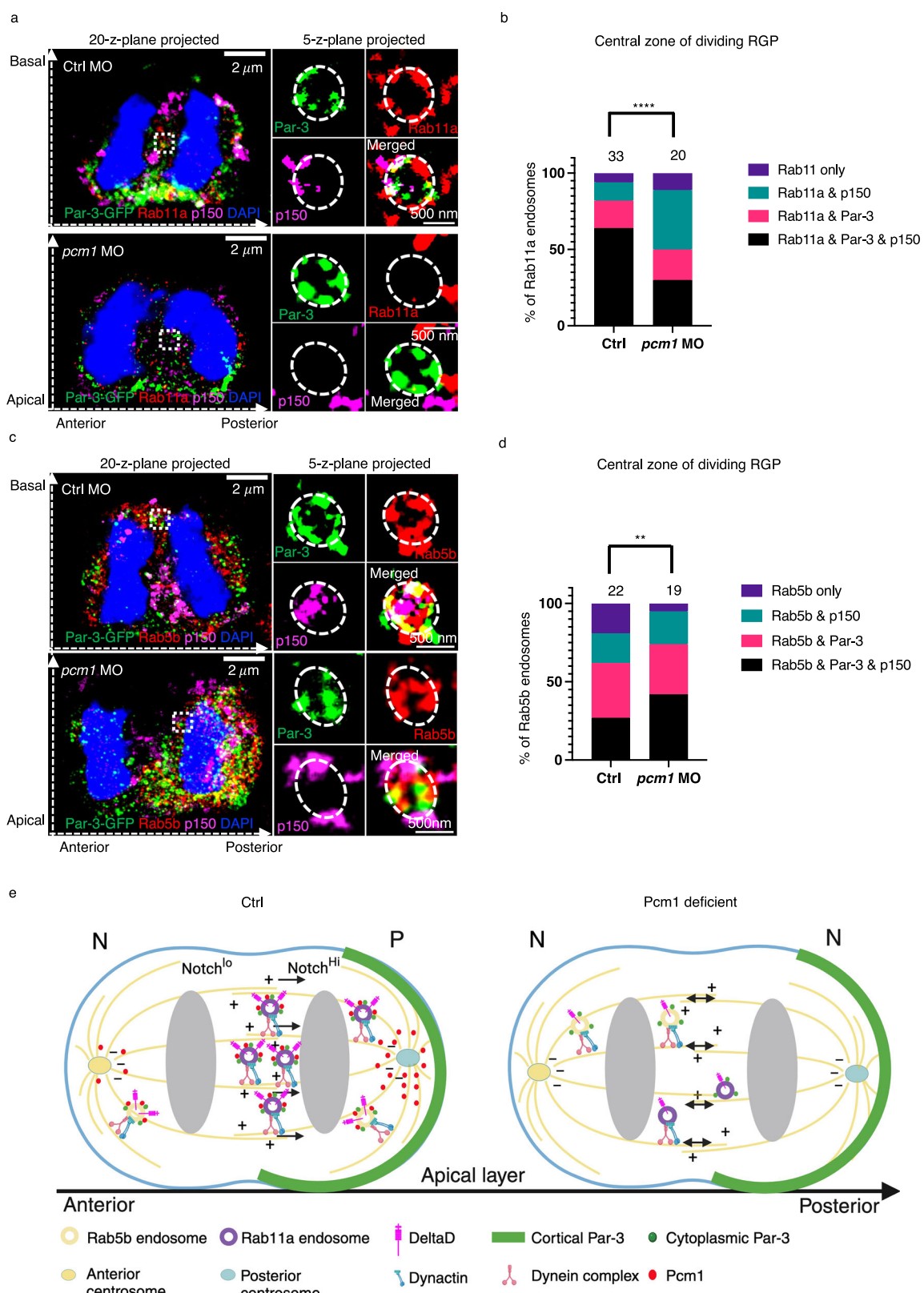

**e**

**Ctrl**                                                    **Pcm1 deficient**

Apical layer

Anterior → Posterior

○ Rab5b endosome    ◯ Rab11a endosome    | DeltaD    ▬ Cortical Par-3    ● Cytoplasmic Par-3

● Anterior centrosome    ● Posterior centrosome    ⚲ Dynactin    人 Dynein complex    ● Pcm1

distribution of PCM1 is observed in most hiPSC-derived mitotic neural progenitors, both in 2D neural rosettes and 3D cortical organoids. These findings suggest a conserved role of PCM1 in regulating endosome dynamics and maintaining progenitor fate in early embryonic neural progenitors across vertebrates.

While it is possible that Pcm1 influences endosomal behavior independently of its asymmetric localization at the posterior mother

centrosome, the coordinated enrichment of Pcm1 around the posterior centrosome during metaphase and the posterior-directed movement of endosomes during anaphase suggest a functional link between these two organelles. In *Drosophila* sensory organ precursor cells, endocytosed Delta traffics through asymmetrically distributed Rab11-positive recycling endosomes, implicating centrosomal guidance in this process[47]. Recycling endosomes interact with mother centriole

**Fig. 7 | Pcm1 is required for trafficking from Rab5 early endosomes to Rab11 recycling endosomes in zebrafish embryonic forebrain mitotic RGPs. a** LR-ExM shows Par-3 with Rab11a and P150 on recycling endosomes. The experiment was repeated three times independently with similar results. **b** Statistics of endosomes with different compositions of Par-3, Rab11a, and/or P150 in the central zone of control and *pcm1*-deficient RGPs. Endosomes were from six RGPs for each group, collected from three independent experiments. **** indicates $p < 0.0001$, $\chi^2$ test (Chi-square = 28.17, df = 3). **c** LR-ExM shows Par-3 with Rab5b and P150 on early endosomes. The experiment was repeated three times independently with similar results. **d** Statistics of endosomes with different compositions of Par-3, Rab5b and/ or P150 in the central zone of control and *pcm1*-deficient RGPs. Endosomes were from six RGPs for each group, collected from three independent experiments. ** indicates p = 0.0086, $\chi^2$ test (Chi-square = 11.66, df = 3). For both (**a**) and (**c**), the left panels show whole cell images (MIP of 20 z-planes); the right panels show enlarged views of endosomes (MIP of 5 z-planes). The z-step is 0.26 μm. Scale bars denote the biological size. **e** Schematic summary of Pcm1's role in macromolecular complex formation and dynamics of Dld endosomes and progenitor fate. N: neuron; P: progenitor. Created in BioRender. Zhao, X. (2025) https://BioRender.com/ps32kwp.

appendages, which are required for their transport to the plasma membrane[48]. Based on these observations, we propose the following model: Pcm1, through its association with the mother centrosome marked by Cep83, becomes enriched around the posterior mother centrosome during metaphase. During anaphase, Pcm1 is deployed to the central spindle and associates with Dld-containing endosomes as well as Par-3/Dynein complexes, directing endosome trafficking toward the posterior pole. Interestingly, in human neural progenitors, we observe that PCM1 and CEP83 occupy distinct centrioles within the mother centrosome. Although the smaller size of zebrafish cells (~5 μm vs ~10 μm for human cells) and the resolution limits of spinning disk confocal microscopy prevent definitive resolution of individual centrioles in zebrafish RGPs, immunofluorescent signals for PCM1 and CEP83 show non-overlapping distributions at the posterior mother centrosome, consistent with findings in human cells. Future studies using high-resolution expansion microscopy, biochemical analysis, targeted genetic perturbation, and dynamic in vivo imaging will be necessary to further test this model.

In vivo time-lapse imaging combined with clonal analysis of progenitor-neuron fate demonstrates that zebrafish Pcm1 is crucial for maintaining progenitor fate by regulating the asymmetric segregation of Dld-containing endosomes. This raises an important question about Pcm1's role in mammalian neurogenesis. Mammalian brains are larger than zebrafish brains and utilize intermediate progenitor cells (IPCs) to amplify neuronal output. Although developing zebrafish forebrains contain neural progenitors that divide to produce two neurons[6], they lack a sub-ventricular zone (SVZ) structure analogous to mammalian brains. Disruption of Pcm1 in mice impairs interkinetic nuclear migration in the ventricular zone and results in neuronal overproduction at the expense of neural progenitors, echoing findings in zebrafish[37]. As this study focuses on ventricular RGPs in zebrafish and early human cortical organoids, investigating Pcm1's role in mammalian IPCs remains an important direction for future research.

Previous studies have identified recycling endosomes near the pericentrosomal region[47], where they closely interact with the appendages of the mother centriole[49]. These observations provide a structural basis for coordinated asymmetry between centrosomes and endosomes during ACD. In this study, we identify Pcm1 as a key regulator of this coordination. Our findings suggest that Pcm1 plays a pivotal role in orchestrating intracellular asymmetry across cortical domains, centrosomes, and endosomes to preserve RGP progenitor fate.

While asymmetry of Notch signaling is known to confer distinct daughter cell fates, the impact of asymmetric centrosome inheritance on cell fate has remained unclear. Our findings reveal crosstalk mediated by Pcm1 between centrosomes and endosomes, by which centrosome asymmetry influences daughter cell fates during ACD. This occurs via the regulation of polarized endosome dynamics and the asymmetric Notch signaling.

In the developing zebrafish forebrain, mitotic RGPs inherit apical-basal polarity from their neuroepithelial progenitors, with Par-3 localized to the apical domain. However, the mitotic spindle in most RGPs is positioned such that the cleavage plane bisects the apical domain. This division orientation necessitates a new axis of asymmetry in mitotic RGPs, with cortical Par-3, centrosomal Pcm1, and Dld endosomes polarized along the A-P axis. Pcm1, Par-3, and dynein components are all localized on Dld endosomes, where Pcm1 facilitates trafficking from early to recycling endosomes and promotes co-localization of Par-3 and dynein. Identifying the extrinsic signals that polarize RGPs along the A-P axis and uncovering additional factors that mediate communication between cellular compartments remain key questions for future research.

## Methods

### Zebrafish strains and maintenance
Wild-type embryos were obtained from natural spawning of AB adults maintained according to established protocols[50]. AB adults' pairs at one year old or younger were used for crossing. Embryos were raised at 28.5 °C in 0.3x Danieau's embryo medium (30x Danieau's embryo medium contains 1740 mM NaCl, 21 mM KCl, 12 mM MgSO4•7H2O, 18 mM Ca(NO$_3$)$_2$, 150 mM HEPES buffer). Embryonic ages were described as hours post-fertilization (hpf). To prevent pigment formation, 0.003% Phenylthiourea (PTU) was added to the medium cultured with 24 hpf embryos. The following zebrafish mutants and transgenic lines were used: Tg [ef1α:Myr-Tdtomato] and Tg [HuC:GFP][6,12]. All animal experiments were approved by the Institutional Animal Care and Use Committee (IACUC protocol no. AN206836) at the University of California, San Francisco, USA.

### Generation of *pcm1* knockout zebrafish
nls-zCas9-nls RNA was in vitro transcribed from a plasmid (a gift from Wenbiao Chen)[51], and injected with two sgRNAs (*TCC*-ATTCACTTA-GAGACCAGACCC, Exon 8; *CCT*-CCAATAATAGAGATGGCCGC, Exon 11) into one-cell stage zebrafish embryos. The PCR primers for genotyping are 5'- TTG ACT CGC CTG TAA CTT GTT G-3' and 5'- CAA TGA GGT TAG TGT GGA ATC C-3', which flanked the regions from Exon 8 to Exon 15 of the zebrafish *pcm1* gene. The sequencing primers used are 5'- GGA AGC TGA AGG AGG TGC ACA A 3' and 5'- TGG TGC AGG TGA TAT TCT AGT CA-3'. The F2 generations of homozygous male and female *pcm1* knockout adult fish were used to generate the maternal zygotic *pcm1* KO.

### Primary antibodies
Rabbit anti-γ-tubulin [Sigma Cat# T5192, Research Resource Identifier (RRID): AB_261690, 1:500 for immunostaining], Mouse anti-γ-tubulin (Sigma-Aldrich Cat# T6557, RRID:AB_477584, 1:1000 for immunostaining), Rabbit anti-CEP83 (Sigma-Aldrich Cat# HPA038161, RRID: AB_10674547, 1:500 for immunostaining), Mouse anti-eGFP (Thermo Fisher Scientific Cat# MA1-952, RRID:AB_889471, 1:500 for immunostaining), Rab5b antibody (rabbit polyclonal, Invitrogen Cat# PA5-44574, RRID: AB_2608403, 1:500 for immunostaining), Rab11a antibody (rabbit polyclonal, Thermofisher PO# 715300, RRID: AB_2533987, 1:500 for immunostaining), Chicken anti-NESTIN (NOVUS biotechne, NB100-1604, 1:500 for immunostaining, Rabbit anti-PAX6 (Biolegend, 901901, 1:250 for immunostaining), Mouse anti-PAX6 (BD Biosciences, 562249,1:500 for immunostaining), Rabbit Anti-TBR2 (Abcam, ab23345; RRID, AB_778267, 1:200). Anti-HuC/D (mouse monoclonal, Thermofisher PO# A-21271, RRID: AB_221448, 1:500 for

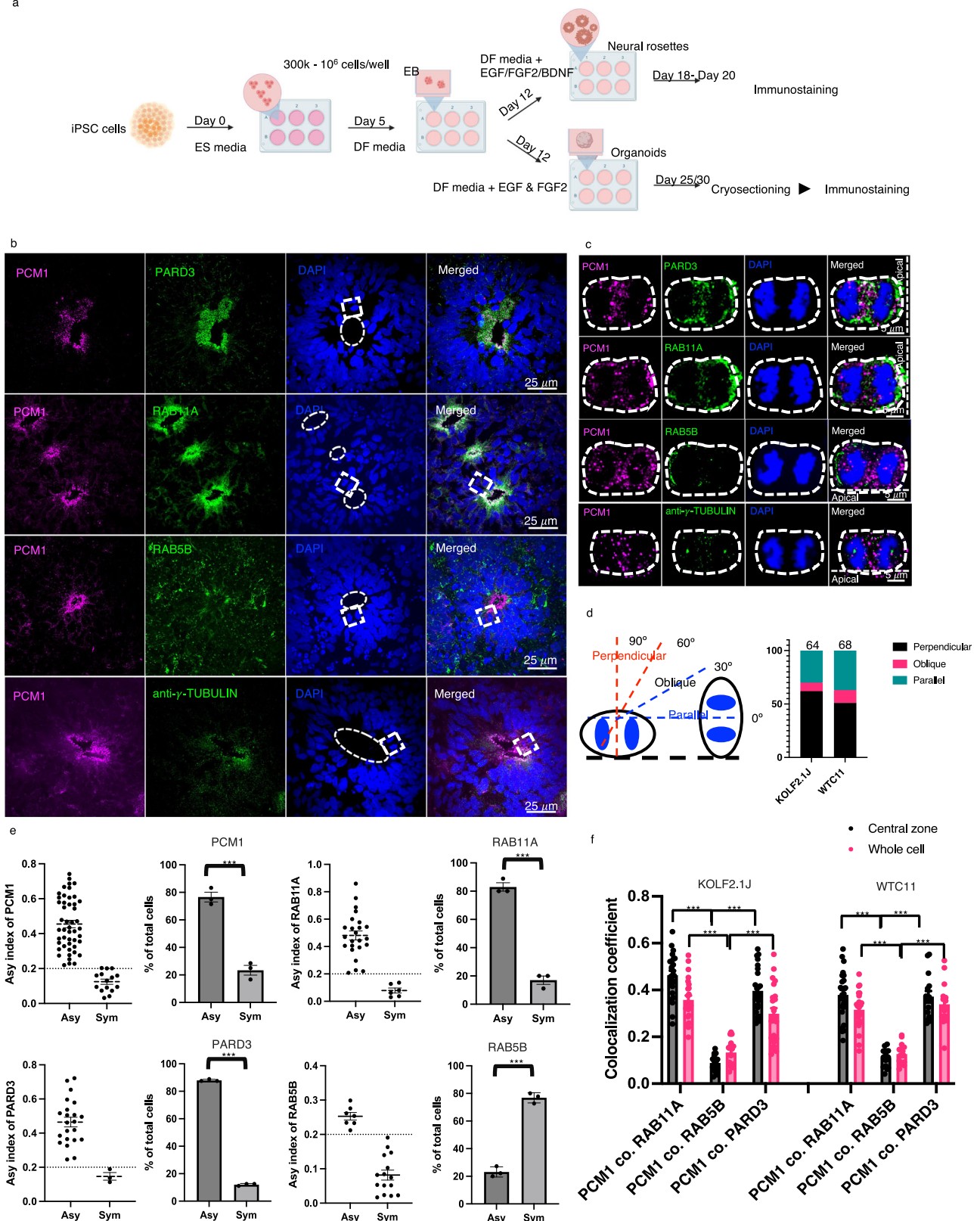

immunostaining), Mouse anti-Dld (Abcam, ab73331; RRID, catalog number: AB_1268496; lot GR115501-3, 1:200 dilution for immunostaining); Chicken anti-GFP (Abcam, catalog number: ab13970; RRID:AB_300798, lot GR3190550-20, 1:500 dilution for immunostaining); Rabbit anti-Par-3 (Millipore 07-330; RRID:AB_2101325; lot 3322358,

1:500 for immunostaining); Guinea pig anti-DLIC1-Cter (a gift from Dr. T. Uemura, 1:100 for immunostaining)[52]; Rabbit anti-Pcm1 (Rabbit antibodies raised against C-terminus of human PCM-1 comprising nucleotides 4993–6095, 1:200 for immunostaining) was provided by Dr. Merdes[26].

**Fig. 8 | PCM1 asymmetry and colocalization with PARD3 and RAB11A in mitotic neural progenitors of human forebrain organoids. a** Experimental pipeline of the derivation of forebrain organoids and neural rosettes from hiPSC lines. Created in BioRender. Zhao, X. (2025) https://BioRender.com/ps32kwp. **b** Detection of PCM1, PARD3, RAB11A, and RAB5B in mitotic neural progenitor cells (NPCs) of forebrain organoid cryosections (14 μm thickness). The ventricular zone (VZ)-like regions are marked with dashed circles. MIP of 20 z-planes is shown. Z-step is 0.26 μm. Scale bar, 25 μm. Three independent repeats showed similar results. **c** Dividing NPCs (marked with dashed rectangles in (**b**)). MIP of 10 z-planes is shown. Z-step is 0.26 μm. Scale bar, 5 μm. White dashed lines in the right panel demarcate the VZ surface. Three independent repeats showed similar results. **d** Statistics of division orientation of NPCs. The cartoon represents three main types of division orientations according to the angles between the cleavage plane and the VZ apical surface. **e** Statistics of PCM1, RAB11A, PARD3 and RAB5B distribution in anaphase/

telophase NPCs. In forebrain organoids derived from KOLF2.1 J hiPSC lines, PCM1, RAB11A, and PARD3 were asymmetrically distributed in most NPCs. In contrast, RAB5B was symmetrically distributed in most NPCs. The asymmetric index of each cell is shown. For PCM1, $n = 48$ (Asy), $n = 15$ (Sym); For RAB11a, $n = 24$ (Asy), $n = 6$ (Sym); For PARD3 $n = 22$ (Asy), $n = 3$ (Sym); For RAB5B, $n = 7$ (Asy), $n = 15$ (Sym). All cells are randomly collected from three independent experiments ($n = 3$). *** indicates $p < 0.0001$, two-tailed unpaired t-test. Error bars indicate SEM in each panel. Center for the error bars is the median. **f** Statistics of colocalization coefficients of PCM1 with PARD3, RAB11A and RAB5B in the anaphase/telophase NPCs of forebrain organoids. *** $p < 0.001$, unpaired t-test. Error bars indicate SEM. For KOLF2.1 J NPCs, $n = 24$ for PCM1 co. RAB11A.; $n = 25$ for PCM1 co. PARD3; $n = 16$ for PCM1 co. RAB5B. For WTC11 NPCs, $n = 24$ for PCM1 co. RAB11A.; $n = 14$ for PCM1 co. PARD3; $n = 15$ for PCM1 co. RAB5B.

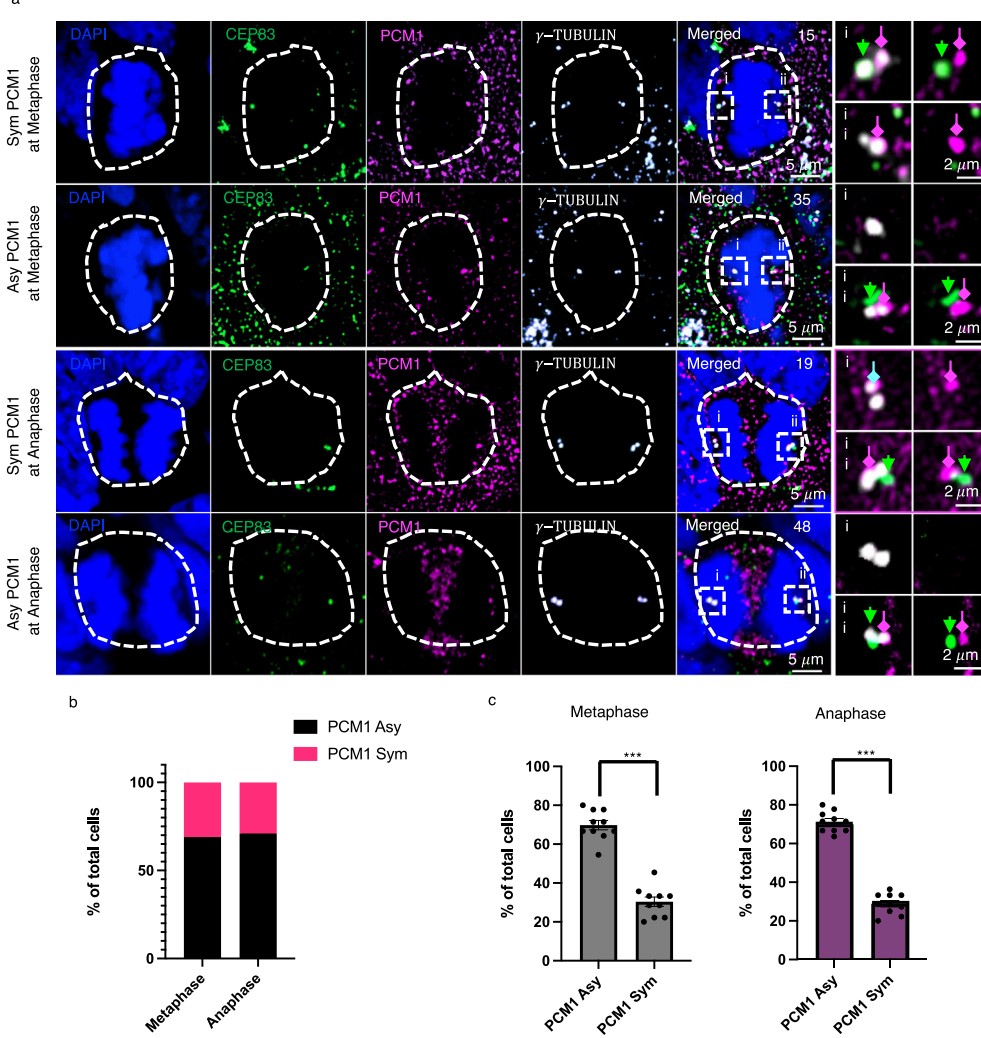

**Fig. 9 | PCM1 asymmetry and colocalization with CEP83 on the mother centrosome in mitotic neural progenitors of human brain organoids. a** PCM1, CEP83 and γ-tubulin in mitotic neural progenitor cells (NPCs) of KOLF2.1 J hiPSC-derived forebrain organoids. The mitotic NPCs with symmetric PCM1 and asymmetric PCM1 distribution at metaphase and anaphase were shown in each row. White dashed line rectangle outlined the pair of centrosomes of each NPC, and enlarged views were shown in i & ii. Each panel is MIP of 10 z-planes. Z-step is 0.26 μm. For the panel showing the whole cell, scale bars indicate 5 μm. In the enlarged view of centrosomes (i & ii), scale bars indicate 2 μm. Green arrows

indicate the mother centrioles with CEP83. Magenta arrows indicate the daughter centriole with PCM1. Three independent experiments with similar results. **b** Statistic of NPCs with asymmetric and symmetric PCM1 on centrosomes. NPCs at metaphase and anaphase show a similar ratio of PCM1 asymmetry on centrosomes. **c** Statistic of NPCs with PCM1 asymmetry on centrosomes collect from 10 forebrain organoids. There are significantly more NPCs showing asymmetric PCM1 with centrosomes than symmetric PCM1 at metaphase and anaphase. *** indicates $p < 0.0001$, two-tailed unpaired t-test, $n = 10$. Error bars indicate SEM.

## Generation of chicken anti-PCM1 antibody

The human PCM1 C-terminal fragment was cloned into pGEX-2TK GST fusion vector (GE Healthcare Life Sciences) using the primers: 5'- ATA GGA TCC CTG AAA GAC TGT GGA GAA GAT C-3' and 5'- ATG AAT TCG ATG TCT TCA GAG GCT CAT C-3' to add the BamHI and EcoRI restriction enzyme sites on both sides of the coding sequence. The plasmids were transformed into *Escherichia coli* XL-1 blue strain by electroporation (Bio-Rad) for PCM1-c-term-GST recombinant expression. The recombinant protein from the cell lysate was purified using Glutathione Sepharose® 4B GST affinity columns (Millipore Sigma, GE17-0756-05). The purified PCM1-GST fusion protein was analyzed on SDS-PAGE stained with Coomassie Blue (Invitrogen). It was used as an antigen for generating IgY anti-PCM1 antibodies produced in the immunized egg yolk by Aves Labs (https://www.aveslabs.com/). The IgY antibodies in the immunized egg yolk were first purified by ammonium sulfate precipitation[53], followed by affinity purification on the affinity column (Pierce NHS-Activated Agarose, Thermo Scientific, PO# 26197; Pierce Centrifuge Columns, 10 mL, gravity or centrifuge compatible, Thermo Scientific PO# 88988) loaded with PCM1-c-term-GST fusion protein. The purified Chicken IgY anti-PCM1 antibodies were diluted to 1 mg/ml aliquots containing 0.01% NaN3 and stored at −80 °C. The working dilution is 1:100 for fluorescent immunostaining and 1:500 for western blotting.

## Secondary antibodies and DNA labeling

Alexa®-conjugated goat anti-rabbit (Alexa 568, Invitrogen, catalog number: A11011, RRID. AB_143157, lot 792518, 1:2000 dilution); Goat anti-chicken (Alexa 488, Invitrogen, catalog number: A11039, RRID:AB_142924, lot 2020124, 1:2000 dilution); Goat anti-mouse (Alexa 488, Invitrogen, catalog number: A11002, RRID:AB_2534070, lot 1786359, 1:2,000 dilution); Goat anti-guinea pig (Alexa 488, Invitrogen, catalog number: A11073, RRID:AB_2534117, lot 46214 A, 1:2000 dilution) or Donkey anti-guinea pig (Alexa 647, Jackson Labs, catalog number: 706-605-148, RRID:AB_2340476, lot 102649-478, 1:2000 dilution); Anti-Mouse-IgG-Atto647N (Sigma-Aldrich, catalog number: 50185, 1 mg/mL); DAPI solution (1 mg/mL) (Thermo ScientificTM, catalog number: 62248, 1:2000 dilution).

## Immunocytochemistry

Samples were first washed and pre-incubated in PBS containing 0.1% Tween 20 or 0.25% Triton X-100 (PBS-T; pH 7.4) with 1% DMSO and 5% natural goat serum at 4 °C overnight or longer. They were then incubated with primary antibodies in the preincubation solution (PBS-T with 5% natural goat serum) overnight at 4 °C. According to the primary antibodies applied, the samples were then washed thoroughly with PBS-T five times × 10 min each time, followed by incubation in Alexa-conjugated goat anti-rabbit (Alexa 568), goat anti-chicken (Alexa 488), goat anti-mouse (Alexa 647), or goat anti-guinea pig (Alexa 647) secondary antibodies (diluted 1:1000) in the preincubation solution for over 2 h at room temperature or overnight at 4 °C. The samples were washed with PBS-T twice for 10 min each, thrice with PBS for 10 min each, and once with 50% glycerol in PBS for 1 h, followed by infiltration overnight in 80% glycerol/PBS before imaging. Imaging was done using a confocal microscope (Nikon CSU-W1 Spinning Disk/ confocal microscopy) with a 100 × oil immersion objective. The z-step of the imaging stack is 0.26 μm.

## Label retention expansion microscopy (LR-ExM)

LR-ExM was carried out according to the established protocol[38]. In brief, after primary antibody incubation as described above, the cryosections of 1 dpf zebrafish embryos (10 μm thickness) were incubated with trifunctional linkers, anti-mouse Atto647N (1:500), NHS-MA-Biotin conjugated anti-Rabbit IgG or anti-Chicken IgY (2 mg/ml) and NHS-MA-DIG conjugated anti-Rabbit or anti-Guinea Pig IgG (2 mg/ml), in the blocking buffer overnight at 4 °C in the dark. After

washing in 1 x PBS 4 times, 5 min each in the dark, add 40 μL of freshly prepared gelation solution (190 μL of Monomer solution with 5 μL of 8% TEMED and 5 μL 8% APS) to cover the whole tissue sample on the glass. The gelation solution was freshly prepared by deoxygenizing the gel monomer solution (Sodium acrylate 0.86 g/mL, Acrylamide 0.25 g/mL, N, N'-Methylenebis-acrylamide 0.015 g/mL, Sodium chloride 1.17 g/mL in 1× PBS) using a vacuum pump for over 15 min, before adding TEMED and APS, to enhance the effects of trifunctional linkers. Protect the sections from light, incubate in a humidity chamber, and allow to undergo gelation at 37 °C for 1 h. Incubate the gel-embedded samples in the digestion buffer (8 units/mL of Proteinase K in 50 mM Tris, pH 8.0, 1 mM EDTA, 1 M NaCl, 0.5% of Triton X-100) on the slides for 4 h at 37 °C or overnight at room temperature. At least 10-fold excess volume of digestion buffer was used (>5mL for each slide). Tissue sections embedded in gels would slide off the glass surface after sufficient digestion incubation. Wash the gel-embedded samples with excess volume of 150 mM NaCl in 6-well plates, with at least 5 mL in each well, for 4 times, 20–30 min each time. After washing off the digestion buffer, incubate the gel samples in the post-digestion staining buffer (10 mM HEPES, 150 mM NaCl in MilliQ water, pH 7.5) with 3–5 μM Alexa Fluor® 488-Streptavidin, 3–5 μM goat anti-Digoxigenin/Digoxin Dylight® 594, and DAPI (1:1,000) for 24 h at 4 °C in the dark. Wash the gel embedded samples 4 to 5 times with Milli-Q water (30 min for each wash) at 4 °C in the dark. The samples expand to approximately four times their original size in the last washing and are then ready for imaging with Nikon CSU-W1 Spinning Disk/confocal microscopy by using 60 × water immersion objective. The z-step of the imaging stack is 0.26 μm.

## Secondary antibodies used for tri-functional linker conjugation of LR-ExM

Goat anti-Guinea Pig IgG (H + L) unconjugated secondary antibody (Invitrogen, catalog number: A18771, RRID:AB_2535548); Goat anti-Rabbit IgG (H + L) Cross-Adsorbed unconjugated secondary antibody (Invitrogen, catalog number: 31212, RRID:AB_228335); Goat anti-Chicken IgY (H + L) unconjugated secondary antibody (Invitrogen, catalog number: A16056, RRID:AB_2534729); NHS-MA-Biotin conjugated anti-Chicken IgY, NHS-MA-Biotin conjugated anti-Rabbit IgG, NHS-MA-DIG conjugated anti-Rabbit IgG and NHS-MA-DIG conjugated anti-Guinea Pig IgG (Gift from Dr. Shi Lab)[12].

## Cryo-sectioning

24-hpf embryos were fixed overnight at 4 °C in phosphate-buffered saline (PBS) buffer with 4% paraformaldehyde (PFA). Fixed embryos were washed by 1 x PBS for two times, 5 min each time, and incubated in PBS buffer containing 30% sucrose in the falcon tube overnight at 4 °C. After embryos sank to the bottom of the Falcon tube, they were then transferred to plastic molds. The sucrose buffer was removed before adding OCT (Tissue-Tek) into the mold to cover the embryos. After orienting the embryos to proper positions in the mold, the block was frozen on dry ice. Blocks can be stored at −80 °C up to several months. Frozen blocks were then cut into 10 μm or 20 μm sections on a Cryostat (Leica) and mounted on Superfrost Plus slides (Thermo Fisher Scientific). The slides were dried at room temperature for 2 - 3 h and then stored at −80 °C until use.

## 5-ethynyl-2'-deoxyuridine (EdU) labeling

Click-iT™ EdU Cell Proliferation Kit for Imaging Alexa Fluor™ 555 (Invitrogen™; Catalog number: C10338) was used for whole mount labeling of cell proliferation in 20-22 hpf zebrafish embryos. EdU concentration was diluted 1:1000 (20 μM) in the working solution. For the EdU pulse-chase experiment, 20 hpf embryos were pulsed in 1 x EdU working solution in the 6-well plate (3 mL/ well) for 2 h at 37 °C. Following the pulse, embryos were immediately rinsed in the new well with Ringer's solution three times (5 min each). The EdU pulse-labeled

embryos were moved to a new well with egg water and incubated at 28 °C till the next morning. After the EdU pulse-chase, the embryos were fixed with 4% PFA and processed for EdU detection according to the manufacturer's protocol. Other antibody staining was applied thereafter, while the samples were protected from light during immuno-staining. For EdU pulse labeling, 22 hpf embryos were incubated with EdU contained working solution 2 h at 28 °C, followed with fixation and EdU staining directly.

**In vivo co-immunoprecipitation (co-IP) and Western blotting**

~ 200 embryos of 1dpf were placed in 1 mL modified Ringer's solution (116 mM NaCl, 3 mM KCl, 4 mM CaCl2, 5 mM HEPES; pH 7.5) containing Protease Inhibitor (Roche, Cat. No. 04693132001) and Phosstop (Roche, Cat. No. 04906845001). After incubation at room temperature on the shaker for 10 min, the supernatant was removed, and 0.3 ml lysis buffer (50 mM Tris, pH 7.5, 150 mM NaCl, 1 mM EDTA, 10% glycerol, 1% Triton X-100) with Protease Inhibitor and Phosstop was added. The tube was kept on ice, and the embryos were homogenized manually by pumping through the syringe with 22-gauge needles for 30–40 times. After homogenization, the tube was placed on the shaker in ice for over 30 min. Then the tube was centrifuged for 30 min at 10,000 x g at 4 °C, and the supernatant was transferred to a fresh tube for the following experiment.

SureBeads™ Protein A Magnetic Beads were used for co-IP with anti-Pcm1 antibody (1:20 dilution). The beads were thoroughly resuspended before use; 150 μl beads were transferred to 1.5 ml tubes. After washing with 1 ml PBS-T (PBS + 0.1% Tween 20), 200 μl PBS-T with 5 μg antibody was added to resuspend the beads. After rotating for 2 h at room temperature (RT), magnetized beads were collected, and the supernatant was discarded. The antibody-conjugated beads were then washed with 1 ml PBS-T for three times and added to the 300 μl embryonic lysate prepared above. After incubation on the rotor at 4 °C for overnight, magnetized beads were collected and supernatant was discarded, followed by five times wash with 1 ml PBS-T. Before the last magnetization, the resuspended beads were transferred to a new tube and centrifuged for 30 s. Beads were then collected on a magnet and residual buffer was aspirated from the tube. 60 μl 1x Laemmli buffer (diluted by mixing same volume of 1x PBS with 2x Laemmli buffer, Bio-Rad, Catalog number: #1610737EDU) was added to the beads and incubated for 10 min at 70 °C. Eluents were then transferred to a new tube. Final collected samples were boiled at 99 °C for 10 min before running SDS-PAGE, or storage at −20 °C.

For Western blotting, ~ 20 embryos of 28 hpf were dechorionated. After removing the yolk with a syringe needle, embryos were washed once with PBS buffer, followed by homogenization in 80 μl SDS sample buffer, the recipe of which can be found in THE ZEBRAFISH BOOK[54]. Pipetting over 50 times was applied till the lysate is not stringy anymore. The sample tube was transferred into a water bath and boiled at 99 °C for 5 - 10 mins, followed by centrifugation for 10 min at 10,000 rcf. The supernatant was transferred to a new Eppendorf tube, which can be stored at −80 °C or used for western blotting immediately. 15 μl of homogenized lysate was used for SDS–polyacrylamide gel electrophoresis (Mini-PROTEAN® TGX™ Precast Gels, 10 wells, Bio-Rad). Proteins were transferred to a Hybond nitrocellulose membrane (Thermo Fisher Scientific, Catalog number: 88025) by a semi-dry blotting technique with Trans-Blot® Turbo™ Transfer System (Bio-Rad) and the membrane was incubated with appropriate primary antibody diluted in 5% dried milk in PBST (1 x PBS, 0.05% Tween-20) overnight at 4 °C (for anti-PCM1, anti-Rab5b, anti-Rab11a, anti-Dld antibody, the dilution is 1:1000; for anti-Par3 and anti-Dlic1, the dilution is 1:500) followed by incubation with corresponding HRP-conjugated secondary antibodies in 2% dried milk in PBST. After the HRP-conjugated secondary antibody incubation, the membrane was rinsed in 1 x PBST once and five times in 1 x PBS, 5-min each. The membrane was then developed using the SuperSignal West Dura Extended Duration Substrate (Thermo Fisher Scientific, Catalog number: 34075) and visualized with the LI-COR Western blotting detection system (LI-COR Biosciences).

**Bulk RNA-Seq and analyses**

Wild-type AB strain zebrafish (*Danio rerio*) were maintained at 28.5 °C on a 14 h light/10 h dark cycle. At the time of mating, breeding males and females were separated in the crossing tank overnight before letting them spawn naturally in the morning to synchronize developmental stages. Fertilized eggs were grown at 28.5 °C for 24 h. Each sample was pooled with 12 embryos (in total ~ 2–3 mg) from the same crossing pair. The pcm1 KO mutants with typical morphological defects as pcm1 MO morphants were selected. Three different samples from three independent experiments were used for both WT and KO group of samples. Each pooled sample of embryos were washed by 1 ml of PBS at room temperature twice quickly. After removing the liquid from the tube, the sample tubes were frozen in liquid nitrogen immediately. The frozen samples were sent to Genewiz (Genewiz from Azenta Life Sciences) to continue with the standard RNA-Seq, Illumina HiSeq, 2 × 150 bp configuration, single index, ~350 M raw paired-end reads per lane, Quality Guarantee is ≥80% of bases ≥ Q30. Raw data have been in FASTQ format.

For RNA-seq data analysis, the quality control was first carried out on the raw reads using FASTQ with default parameters, including sequence quality, GC content, adapter content, overrepresented k-mers, and duplicated reads analysis, aiming at detecting sequencing errors, contaminations, and PCR artifacts. The qualified reads were aligned to the reference genome GRCz11.101 by Hisat2, and the transcripts and corresponding read counts were obtained subsequently. The aligned BAM file quality control was achieved using the module bam_stat.py. The count matrix of the transcripts was generated by HTSeq, and the differentially expressed genes (DEGs) were identified by using the module DESeq2. Following the general protocols, a DEG was defined as a gene with log2(FC) > ± 0.5 and p-value < 0.05 between the two groups, *i.e.*, KO vs. WT in this study. The visualization and the analysis of the DEGs were performed via the Pheatmap R package. The molecular functions and the biological pathways of the enriched DEGs were analyzed using the clusterProfiler package[55]. GO term enrichment analysis was performed with the GO cell component sources, the GO molecular functions sources and the GO biological processes sources in the Metascape[56]. Enriched terms ($P < 0.05$, a minimum count of 8, and an enrichment factor of >1.5) were grouped into clusters based on their DE gene similarities and visualized in a network plot using Cytoscape, in which node colors indicate the P value, node sizes indicate DE gene set size, and the connecting edges indicate their DE gene similarities. Processing of RNA-seq data and clustering analysis were performed blindly and unblinded in the following DEG analysis. Heatmaps with hierarchical clustering were made using the heatmap.2 packages with the transformed values in which the expression of each gene was scaled across all three samples in both *pcm1* KO and WT groups (z-scored).

**DNA Plasmids, mRNA synthesis and microinjection**

Plasmid DNAs (pCS2-H2B-mRFP) were extracted from bacterial clones in stock[6]. pCS2-par-3-GFP plasmid was a gift from Dr. J. von Trotha[57]. pCS2-GFP-centrin plasmid was a gift from Dr. W. A. Harris[58]. The full-length of cDNA of zebrafish Pericentriolar Material 1 (pcm1, Gene ID: 321709, Ensembl: ENSDARG000000-62198) were synthesized and sequenced by Genecreate (www.genecreate.com). The pcm1-linker-GFP plasmids was made by cloning pcm1 cDNA into pcs2-GFP plasmid (gift from the Jan Lab) by Gibson assembly. PCR primers for adding linker between *pcm1* cDNA and GFP cDNA sequence were used as follows: GFP-pCS2-Linker-R: CGT GGA CCC GCC ACC ACT CCC GCC ACC AGA TTT GTA TAG TTC ATC CAT; pcm1-Linker-F: TCT GGT GGC GGG AGT GGT GGC GGG TCC ACG ATG GCA ACG GGT GGC ACT; pCS2-PCM1-R: CCG TTG CCA TGG TGG ATC CTG CAA AAA GAA C;

pcm1-pCS2-F: AGG ATC CAC CAT GGC AAC GGG TGG CAC TCC; pcm1-Linker-R: CGT GGA CCC GCC ACC ACT CCC GCC ACC AGA TGC ACT CTG GGC TCC AAT; pCS2-Linker-F: TCT GGT GGC GGG AGT GGT GGC GGG TCC ACG ATG AGT AAA GGA GAA GAA. For par3-GFP, GFP is at the 3′ end of Par-3 protein. For GFP-centrin, GFP is at the 5′ end of Centrin. For pcm1-linker-GFP, GFP is at the 3′ end of Pcm1. The pcs2⁺-CEP83-EGFP plasmid was made by cloning CEP83-EGFP cDNA from pEGFPC1-CEP83 (Addgene, #128874) into the pcs2⁺ plasmid vector using PCR primers: pcs2⁺-XbaI:TAT CTA GAT CAC TCC CCA GGA GAC CCA AGC TCC and pcs2⁺-BamHI: TAG GAT CCG CCA CCA TGG TGA GCA AGG GCG AGG A.

Plasmids (pCS2-H2B-mRFP; pCS2-par-3-GFP; pCS2-GFP-centrin; pCS2-pcm1-linker-GFP) were linearized by the restriction enzyme NotI digestion. NotI-linearized plasmids were purified (QIAquick Gel Extraction Kit), and the 5′-capped mRNAs were synthesized using SP6 mMessenger mMachine kit (Ambion). For mRNA injection, 4 nl mRNAs at 0.5–1 μg/μl were mixed with equal volume of injection buffer containing 0.05% phenyl red and injected into the yolk of a 1–4 cell stage embryos. All injections were done with an injector (WPI PV830 Pneumatic Pico Pump) and a micromanipulator (Narishige, Tokyo, Japan).

## Morpholino oligos and microinjections

Knockdown experiments were carried out using previously characterized translational blocking antisense morpholino oligonucleotides (MOs): *pard3ab/par-3* MO (5′-TCA AAG GCT CCC GTG CTC TGG TGT C-3′)[12]; *pcm1* MO (5′-TGG AGT GCC ACC CGT TGC CAT GAT G-3′)[59]; Standard control MO (5′-CCT CTT ACC TCA GTT ACA ATT TAT A-3′) was used as an injection control. All MOs were ordered from Gene Tools (https://www.gene-tools.com/) and stored at 300 mM in distilled water. For microinjection, ~ 4 nL MO at 100 mM was injected into the yolk of 1–2 cell stage embryos containing 0.05% phenyl red (corresponding to 4 ng/zygote).

## Hybridization Chain Reaction (HCR)

We investigated Her4.1 mRNA transcript expression in the forebrain of 1 dpf larvae by using HCR[60]. By Molecular Instruments, we designed a probe set for targeting Her4.1 mRNA. We applied RNA in situ hybridization according to the Molecular Instruments RNA FISH protocol for zebrafish embryos (MI-Protocol-RNAFISH-Zebrafish-Rev9). Before in situ hybridization, all 1dpf embryos were dechorionated after incubating within 1 mg/ml pronase solution for no longer than 90 s at room temperature, then washed thoroughly with egg water. After finishing the step of amplification, all embryos were incubated with DAPI (1:5000 dilution) for 2 h at room temperature or overnight at 4 °C in the washing buffer (5 × SSCT). After that, all embryos were washed three times according to the protocol before storing at 4 °C protected from light. For imaging, we have mounted the whole embryo in 80% glycerol on the glass slide and used Nikon CSU-W1 Spinning Disk/High Speed Whitefield confocal microscopy.

## In vivo time-lapse imaging, image processing, and analyses

All the confocal imaging stacks were captured by Nikon CSU-W1 Spinning Disk/High Speed Widefield confocal microscopy with Andor Zyla 4.2 sCMOS camera by using 40 × water immersion objective (for live imaging) and 100 x oil immersion objective (for immunofluorescent imaging, OILCL30 - Very Low Autofluorescence Immersion Oil, $n = 1.518$, Cargille Type LDF, Thorlabs) with Micro-Manager 2.0 gamma (μManager, University of California).

For live imaging, the glass-bottom petri dish with embryos was placed on a temperature-controlled stage set at 28.5 °C. For in vivo time-lapse imaging of internalized Dld, centrin-GFP or pcm1-GFP in dividing RGPs of *Tg [ef1a:Myr-Tdtomato]*, z-stacks with 20 to 40 z-planes were acquired consecutively at a 1-μm z-step for each embryonic forebrain region. The exposure time for each fluorescent channel was set at 100 ms by choosing the sequential channel scanning

mode for each z-plane. The interval between each z-stack ranged from 12 to 30 s, depending on the z-stack settings of the samples. Usually, 80 volumes of z-stacks were captured for each time-lapse imaging, and the duration spanned around 20–30 min. The sequential scanning mode, which is the one we used, sets 100 ms exposure per z-plane (z-step = 1 μm). This minimized the scanning period of each channel to be only 3 s for the whole volume of 30 μm. For triple channel scanning, the time interval between each time frame is ~12 s. The time between each channel is, therefore, ~4 ss.

For imaging HuC-GFP in paired daughter cells, H2B-mRFP mRNA was injected into a single cell of *Tg [HuC-GFP]* embryos at 16–32 cell-stage. Z-stacks with 50 to 60 z-planes were acquired consecutively with a 1-μm z-step for each volume. The scanning interval between volumes of z-stacks was 6 min. The exposure time for each channel was set at 100 ms for each z-plane as described above. For each embryo, we have taken about 200 volumes of z-stacks lasting ~20 h. In figure panels and movie frames, maximum intensity projections of 5 to 10 z-planes with 1-mm z-step size were shown, representing the approximate size of RGP and covering both daughters throughout the time-lapsing frames. As for the sequential scanning mode, the frame of different channels might shift a bit due to the spontaneous movement of zebrafish embryos, which was later motion corrected during analysis.

After imaging acquisition, Fiji[61] was used for image processing. For the measurement of fluorescent intensity of each antibody staining, maximum intensity projection (MIP) of 8–10 z-planes (0.26 μm z-step) was applied to the image stacks to cover the z-planes in the middle of the whole RGP. The DAPI staining of each sample was used for the normalization of staining intensities of the other channels of the same sample. The normalized immunofluorescent intensity of the same antibody was use for the comparison among different samples. For the measurement of endosomal staining area, MIPs of endosomes were used. After switching to grayscale style for each channel, the threshold was set 100–255. The endosomal surface area was measured by Par-3 and Dlic1 labeling separately if both are labeled.

For cell nuclei counting, the Fiji plugin for StarDist was used as described[62]. For the colocalization analyses of Par-3, Pcm1, Rab11 and Dld, the area (50 × 100 pixel, 0.126 μm/pixel) in between the mitotic nucleus was chosen for applying JACoP analyses with Fiji[38]. In brief, we measured the colocalization between every two fluorescent channels in the form of Manders' Coefficients (i.e., the proportion of each fluorescence colocalized with the other) by using the JACoP plugin in Image J[63]. For removing nonspecific signals, the threshold of each channel was set in JACoP using a blank area on the image (i.e., without tissue samples) at first. Then we ran Costes' automatic threshold, which is an algorithm for identifying and removing noise using scatter plotting of randomized images generated from the image under analysis[64]. For tracking internalized DLD particles in dividing RGPs and the determination of distribution at different mitotic cell phases, Trackmate has been applied for each group of the live-imaging dataset[12]. Kymopy has been used for generating the kymograph of internalized Dld dot[65]. The spatial registration of each time frame was done by adopting the center point between two centrosomes as the center of dividing RGPs. The two centrosomes labeled with Cen-GFP were used to define the anterior-posterior axis: the anterior centrosome was given the coordinate 0, and the posterior centrosome 1. Each DLD endosome in the RGP cell was then projected onto this axis to obtain its relative distance (value between 0 and 1) at each time frame. On kymograph images, the grayscale value of each pixel indicates the probability of all tracked Dld endosomes at the corresponding location at each time frame. The relative distances of all tracked Dld endosomes were then used for calculating distance and velocity. The temporal registration among different time-lapsing image dataset was done by adopting the anaphase with the first appearance of cleavage furrow to be T = 0 min (ref. [12].). The 20-time frames before and after T = 0 min were used for kymograph analyses afterwards. The asymmetry index

of fluorescent immunostaining and live labeling within two newly formed daughter cells was calculated[12]. The total fluorescent intensity of antibody immunostaining in both parts of the same RGPs was measured by FIJI, and the fluorescent background was set to 0 before the measurement. The asymmetric expression was determined when the total fluorescent intensity of antibody immunostaining in one part is 50% more than the other part (posterior asymmetric when the asymmetric index >0.2 or anterior asymmetric when the asymmetric index < −0.2). If less than 50%, the expression was determined as symmetric (-0.2 <asymmetric index <0.2). The asymmetric index of perfect symmetry is "0", and "1" or "−1" indicates absolute asymmetry (posterior or anterior, respectively).

For LR-ExM images shown in figure panels, Aydin denoising was applied to process the whole stacks of z-planes with all four channels respectively (https://royerlab.github.io/aydin/, https://doi.org/10.5281/zenodo.6612581). The "Butterworth" denoising algorithm was used for removing the immunostaining background and noise caused by unspecific or unstable trifunctional linker signals without removing specific signals. Aydin Butterworth denoising algorithm calibrates an optimal Butterworth filter for the given image and its spectrum with the help of self-supervised Noise2Self loss (https://github.com/czbiohub-sf/noise2self). The low-pass nature of Butterworth filters enables a successful removal of high-frequency noise components without causing any major changes on most of the given image spectrum. Aydin implementation, by default, calibrates the optimal Butterworth filter for each image axis to prevent artifacts that might occur due to spectrum variances across different axes. The difference between raw images and denoised images can be found in Supplementary Fig. 5. For the normalization of fluorescent signals of Dld on endosomes, we used two alternative methods (both give similar results). In one method, the DAPI signal was used for normalization[66,67]. In the other method, the percentage of endosome surface area (demarcated by either Par-3 or Dlic1) occupied by the Dld signal was used as the primary quantification metric. This approach provides a channel-independent measurement that is not affected by laser power settings or absolute fluorescence intensities.

### Human-induced pluripotent stem cell lines
Human induced pluripotent stem cell lines (hiPSC) WTC11 (Allen Institute, RRID: CVCL_Y803, derived from male fibroblast of skin) and KOLF2.1J (The Jackson Laboratory, RRID: CVCL_B5P3, derived from male fibroblast of skin) were used. All pluripotent stem cell lines were cultured at 37 °C on Matrigel (Corning Catalog No. 354277) in Gibco™ StemFlex™ Medium (Thermofisher, Catalog No. A3349401) and amplified using Accutase (Stem Cell Technologies, Catalog No. 07920) for passaging. iPSCs were thawed in the presence of Y-27632 Rock Inhibitor (Stem Cell Technologies, Catalog No. 72304), and the culture medium was changed every day.

### Human iPSC-derived neural rosettes and forebrain organoids
For each independent vial of neural rosette culture, one million human iPSCs were thawed and resuspended with 1 mL of Dulbecco's Modified Eagle Medium F12 (DMEM F12) media (Thermofisher, Catalog No. 12634010), then diluted in 2 mL of warm DMEM F12 in a 15 mL Eppendorf conical tube (Eppendorf, Catalog No. 0030122151). Our protocol was adapted from previous reports[41,68]. hiPSCs were collected by centrifugation at 200 g for 5 min at room temperature. After Removal of the supernatant carefully, the cells were resuspended in 2 mL Gibco™ StemFlex™ Medium (Thermofisher, Catalog No. A3349401) with 10 μM Rock Inhibitor (Stem Cell Technologies, Catalog No. 72304). The resuspended cells (normally > 500 k cells) were transferred to a single well on the Matrigel (Corning Catalog No. 354277) pre-coated 6-well plate (Corning Catalog No.3516). The plate was then placed in the incubator at 37 °C and 5% $CO_2$ with saturating humidity. The culture media were changed daily. After incubation for

five days, the cultured cells in the well were washed once with PBS. Then 1 mL Accutase was added to cover the cultured cells for 3–5 min at 37 °C. 5 mL DMEM F12 was added to collect cells by pipetting up and down slowly for ~20 times and transferred to a new 15 mL Eppendorf conical tube. More DMEM F12 media was added to 10 mL total volume. The cells were pelleted at 200 g for 5 min at room temperature. The supernatant was removed and the cells were resuspended in 2 mL of differentiation medium N2B27 [Advanced DMEM F12 added with equal volume of Neurobasal (Life Technologies), supplemented with N2 (Life Technologies), B27 without Vitamin A (Life Technologies), penicillin/streptomycin 1%, β-mercaptoethanol 0.1% (Life Technologies)], plus 10 μM Rock-Inhibitor, 0.2 μM LDN (LDN193189, Selleckchem, Catalog No.501362646), 10 μM SB (SB431542, Selleckchem, Catalog No.101762-616) and 5 ng/mL FGF2 (FGF basic 154aa, human, Peprotech, Catalog No.10771-938).

For forebrain organoid cultures, the resuspended cells from above were transferred into a new ultra-low attachment 6-well plate (Corning, Catalog No.07-200-601) with one million cells per well (Day 0). Then the plate was placed back into the incubator, and the differentiation media were changed every other day. The rock inhibitor was omitted in the differentiation media. The embryoid bodies (EB) floated up in the medium at day 5. At Day 16 of differentiation, the differentiation media was supplemented with EGF (10 ng/mL) and FGF2 (10 ng/mL) to promote proliferation and expansion of early organoids. The organoids were collected after day 25 of differentiation. The forebrain organoids with ventricle-like structures were selected for fixation, cryosection, and immunostaining[41].

For neural rosette cultures, the resuspended cells from above were transferred into a new low-adherent 6-well plate (Fisher Scientific, Catalog No.7200601) with ~300 k cells per well. EBs at day 5 were transferred to a new 6-well plate coated with PDL (Millipore. Catalog. MILL-A-003-E)/Laminin (Gibco. Catalog No.2301715) containing fresh media of N2B27 with 0.2 μM LDN and 10 μM SB. The plate was slowly rotated to spread the EB containing media over the well before placing it back into the incubator. The medium was changed once every other day for another 6–8 days. The forming neural rosettes were detected, usually starting at Day 12 and collected under a stereomicroscope with a mini cell scraper (Biotium, Catalog No.NC0325221). The neural rosettes were then seeded to a new PDL/laminin pre-coated well containing N2B27 media with 10 ng/mL FGF2, 10 ng/mL EGF (Fisher Scientific, Catalog No. GF144), and 10 ng/mL BDNF (Peprotech, Catalog No.450-10). The plate was placed back into the incubator with a change of the media every other day. After culturing 6–8 days, the neural rosettes were fixed with cold 4% PFA for 5–10 min and processed for immunostaining.

### Processing of forebrain organoids for immunostaining
Forebrain organoids ( >1 mm in diameter) were collected, rinsed with PBS, and fixed with 4% PFA for 5 min at 4 °C and rinsed with PBS three times for 5 min. The fixed EBs were incubated in 30% sucrose until saturation and sinking to the bottom, followed by embedding in OCT (Tissue-Tek) -filled plastic molds and storage at −80 °C. Frozen blocks were cut into 14 μm sections on a Cryostat (Leica) and mounted on Superfrost Plus slides (Thermo Fisher Scientific). The slides were dried at room temperature for 2 to 3 h and then stored at −80 °C until use.

### Analysis of protein asymmetric distribution and co-localization
For the live imaging analyses, the total fluorescence intensity (Ints) of Dld (or Pcm1-GFP, Par-3-GFP) in paired daughter cells immediately after abscission (i.e., at telophase of mother RGP division) was measured by Image J and normalized by deducting the background intensity measured from the same view without any overlap with samples. To quantitatively describe the distribution, the normalized ratio of fluorescence between the two newly formed daughter cells was

calculated as follows:

$$X = \frac{\sum_{i=1}^{n}(Ints)Post - \sum_{i=1}^{n}(Ints)Ant}{\sum_{i=1}^{n}(Ints)Post + \sum_{i=1}^{n}(Ints)Ant}$$

$\sum_{i=1}^{n}(Ints)Post$ means total intensity in the posterior daughter cell, and $\sum_{i=1}^{n}(Ints)Ant$ means total intensity in the anterior daughter cell. If X = 0, it indicates absolute symmetry. And "X = 1" or "X = −1" indicates absolute asymmetry (posterior or anterior, respectively). For filtering out potential noise, we defined asymmetry when one is 50% more or less than another[12]. It means that when X ≥ 0.2, Dld endosomes (or Pcm1-GFP) are considered asymmetric with more in the posterior daughter, and when X ≤ −0.2, they are considered asymmetric with more in the anterior. The asymmetry index for Par3-GFP included both membrane and cytoplasmic fluorescence.

For the analysis of asymmetry in fixed staining or LR-ExM samples, we measured anterior and posterior fluorescent intensities by using the cell nucleus as the marker. For the cell at metaphase, we split the whole cell into two parts and measured the intensity of the anterior vs. posterior parts. For the cell at anaphase, we split the mitotic cell into three parts and measured intensity on the anterior and posterior sides without including the central zone.

## Statistics and reproducibility

The number of each independent experiment was provided in the figure legends. For immunocytochemistry experiments, multiple sections from individual brain samples were analyzed. For live imaging, three or more dividing RGPs were analyzed from each embryo, depending on the number of mitotic RGPs that were present in each image stack. For each group of samples, the sample size was determined to be adequate based on the magnitude and consistency of measurable differences between groups. No randomization of samples was performed. Embryos used in the analyses were age-matched between control and experimental conditions, and sex cannot be discerned at these embryonic stages. Investigators were not blinded to genetically or MO microinjection-perturbed conditions during experiments. Data are quantitatively analyzed. Statistical analyses were carried out using Prism 9 version 9.0.0. The mean value with standard error of the mean (S.E.M.) or standard deviation of the mean (S.D) was labeled in the graphs. The two-tailed unpaired t-test and the *Chi-square* analyses were applied to determine significant differences between groups. Statistical significance is determined as follows: ns, P > 0.05; *P ≤ 0.05; **P ≤ 0.01; *** P ≤ 0.001; **** P ≤ 0.0001.

## Reporting summary

Further information on research design is available in the Nature Portfolio Reporting Summary linked to this article.

# Data availability

All data needed to evaluate the conclusions in the paper are present in the paper and/or the Supplementary Materials. The Bulk mRNA-Seq data generated in this study have been deposited in the NCBI SRA database (https://dataview.ncbi.nlm.nih.gov/object/PRJNA1159911). Source Data are provided with this paper. The raw gel data and western blot results generated in this study are provided in the Supplementary Information/Source Data file. Source data are provided with this paper.

# Code availability

Aydin denoising codes are available at https://github.com/royerlab/aydin (DOI: 10.5281/zenodo.5654826). Codes for kymograph annotation are at https://gitlab.com/bio4212310/kymopy (https://doi.org/10.5281/zenodo.17336924).

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

## Acknowledgments

We thank M. Munchua and E. Lee for excellent animal care; B. Lu, M. Nachury, J. Reiter, and Guo laboratory members for helpful discussions; S. Schmid for helpful comments on the manuscript; D. Larsen, K. Herrington, and UCSF Nikon imaging center for assistance with imaging and data analysis; Dr. J. von Trotha for the pCS2-Par-3-GFP plasmid, Dr. W. A. Harris for the pCS2-GFP-centrin plasmid, Dr. A. Merdes for the anti-hPCM1 antibody, Dr. T. Uemura for the anti-DLIC antibody, Dr. Shao and Dr. Gestwicki for the human PCM1 antigen purification. Funding: This project was supported by NIH R01NS120218 and R21 R21NS122053 (to S.G.), the UCSF Mary Anne Koda-Kimble Seed Award for Innovation 2021 (to X.Z.), and Chan Zuckerberg Biohub San Francisco (X.Z. & L.R.).

## Author contributions

X.Z. and S.G. designed the experiments and interpreted the results. X.Z. performed most experiments, Y.W. performed bulk RNA-Seq annotation, contributed to Fig. 6, A.C.S. performed kymograph annotation in Fig. 2 and offered guidance for Aydin denoising that contributed to Figs. 5, 6, X.C. has provided all codes for live tracking of internalized Dld and results annotation in Fig. 2, V.M. prepared hiPSC cultures used in the study. J.G. made the plasmids clones of zebrafish pcm1. X.S. offered material and instruction for expansion microscopy experiments, Z.D. performed CRISPR and assisted full-length zebrafish pcm1 sequencing and clone, L.R. assisted with critical steps in image analysis, and C.J.W. provided reagents. X.Z. and S.G. wrote the manuscript, with the input from all authors.

## Competing interests

The authors declare no competing interests.
