## [Transparent Peer Review file · Nature Communications]

PCM1 coordinates centrosome asymmetry with polarized endosome dynamics to regulate daughter cell fate

Corresponding Author: Dr Xiang Zhao

Version 0:

Reviewer comments:

Reviewer #1

(Remarks to the Author)

Key results:

Zhao et al. aim to demonstrate mechanistically how asymmetric segregation of centrosome-associated proteins during mitosis, specifically PCM1, regulates the fate determination of radial glia progenitor cells. Using different cell- and molecular biological techniques, including live imaging *in vivo*, expansion microscopy, immuno-assays, and RNAseq, they aim to show that PCM1, in conjunction with Par-3 and Dynein, would mediate the dynamics of endosomes containing Notch-ligands, ultimately leading the acquisitions of difference fates by the daughter cells after cell division. Indeed, the roles of asymmetric segregation of centrosomes in (neural) stem cell behaviors remain largely unclear, particularly in mammals. Hence, studies that help clarify the contribution of centrosome inheritance for regulating stem cell systems (here neurogenesis) are essential. Even though the work by the authors has the potential to impact the field significantly, and the data (particularly live imaging) strongly enough support their claims, I recommend addressing the following points before considering the study for publications:

1. The authors mention in the introduction that “how centrosome asymmetry confers daughter cell phase differences at the mechanistic level remains unclear.” The lab of Wieland Huttner has been one of the few addressing the roles of asymmetric inheritance of centrosome-associated structures (i.e., cilia) and mother centrioles in radial glia during mammalian neurogenesis mechanistically. I am surprised these papers (Paridaen, 2013, *Cell*; Wilsch-Bräuninger, 2012, *Development*) were not considered, given the relevance for the current manuscript. The protein compositions of the centrosomes have also been shown to play a pivotal role in regulating self-renewal and differentiation of neural stem cells (Camargo Ortega, 2019, *Nature*; Camargo Ortega and Götz, *TICB*, 2021; O’Neill, *Science*, 2021).
2. The authors comment that the PCM1 is likely enriched around the mother centrioles in metaphase, as the cloud-like distribution around gamma-Tubulin was asymmetric, and this would relate to the role of PCM in ciligenesis. It would be valuable if the authors could indeed demonstrate that PCM1 is enriched in the older centrioles in radial glial cells, e.g., by co-immunostaining with a distal appendage marker (e.g., CEP83). Notice that subdistal appendages of the mother centriole are retracted during mitosis; thus, stainings with SDA markers (e.g., Ninein, CEP170) are not appropriate.
3. Generally, the number of samples for quantification is too low. In Figure 1, the n-sample is mostly below 50 cells. In Figure 1i, 27 cells from 6 embryos were quantified. This is four cells per embryo, which is relatively low for confident statistical analyses.
4. At the end of the first results section, in the last paragraph, the authors claim to uncover “a previously unknown asymmetry of Pcm1 at the centrosomes and its localization in the central zone near Dld endosomes following enrichment in the posterior daughter.” The figures (e.g., Figure 1A) and movies show, however, no clear enrichment at centrosomes in mitosis. They show the presence of Pcm1 protein in centrosomes, as expected, but a significant portion (if not most of it) is not at centrosomes but distributed in the cytoplasm in granules. I would suggest rephrasing this statement or clarifying what they mean by enrichment.
5. In the first paragraph of the second results section, the authors mention that “perpendicular divisions were reduced with a corresponding increase of non-perpendicular divisions...” While this is true for the pcm1 KO animals, the same is not the case for pcm1 morpholinos (Extended data Fig. 2a, 2c). I recommend that the authors address these differences in loss-of-

function results.

6. The authors determined that there is a relative increase in neuronal production in *pcm1*-deficient fish embryos using clonal analysis. This experiment is appropriate for determining the effects of *pcm1* loss-of-function in brain development. To improve the resonance of their discoveries, particularly during mammalian neurogenesis, I strongly recommend analyzing the loss-of-function impact (short and long-term) during murine cortical neurogenesis. This is because, unlike fishes, mammalian brain neurogenesis uses intermediate progenitor cells to amplify the final neuronal output. This is most prominent in gyrencephalic species, where there are several intermediate progenitors. As a *Pcm1*-full knock-out may be lethal, I would suggest employing shRNAs or miRNA to down-regulate *Pcm1* and track the cells of interest. This experiment could be done in cortical organoids, but this system mostly resembles early human neurogenesis.

7. The authors suggest that defective Notch signaling may be behind the *Pcm1* loss-of-function-mediated effects in neurogenesis due to altered *Dld* inheritance. I think this point is crucial, and thus, I would recommend analyzing Notch signaling in control vs. *Pcm1*-LOF conditions using Notch-signaling reporters (e.g., HES-GFP strains).

8. The authors use forebrain organoids to compare some of their observations in fish to a human model. Judging by the stainings in the Extended Data Figure 8A, particularly the pattern of TBR2, the results indicate that the organoids have not developed subventricular zones yet, i.e., they are in a relatively early neurogenic stage. If the authors would like to properly compare the role of asymmetric centrosome inheritance in the neurogenesis of mammals, I would – again – recommend checking in murine embryos (e.g., at E14), ferrets, macaques, or further differentiated cortical organoids (in this case, please show Pax6 and Tbr2 stainings together).

Reviewer #2

(Remarks to the Author)

In the presented study Zhao and colleagues investigate the role of *Pcm1* in regulating asymmetric cell division during zebrafish development.

Using a combination of live cell imaging, developmental biology and high-resolution light microscopy of radial glia progenitors (RGPs) the authors show that *Pcm1* is asymmetrically distributed during asymmetric cell division. Intriguingly, the authors show that *pcm1* itself is required for the polarized trafficking of Notch ligand-containing endosomes, a critical process in asymmetric cell division across various species. As a consequence, *Pcm1* influences cell fate determination post-division. Capitalizing on expansion microscopy, the authors then characterise the contents of the endosomes, suggesting that *pcm1* is needed for the assembly of the machinery present in the compartment. Finally, the authors show that their findings are conserved in humans using human induced pluripotent stem cells. The authors suggest a conserved mechanism by which *Pcm1* coordinates centrosome asymmetry with endosome dynamics to regulate asymmetric cell division and subsequent cell fate.

Overall, the experiments are well-conducted and the results are intriguing. This study provides novel insights into the connection between cytosolic cell fate determinants and the centrosome, a previously underexplored area in asymmetric cell division. While the work is promising, certain conclusions require additional data or analysis. Although I am enthusiastic about the authors' research, I feel they occasionally overinterpret their findings and make excessive claims. Nevertheless, these issues can often be addressed by further analysing existing data already collected by the authors or tempering conclusions. Therefore, I support publication in Nature Communications provided these points are resolved. Here are some remarks highlighting this:

Major points:

1) I am not fully convinced that *pcm1* travels with *Dld*-positive endosomes. The only live-imaging available to me (Fig. 1h and associated movies) shows a rather independent movement of *Pcm1* and *Dld* signals. This is not essential for the manuscript's main claim that *Pcm1* regulates polarized endosome trafficking of *Dld*-positive endosomes. However, the current draft strongly implies that *Pcm1* itself is physically located within the endosomes (based on LR-EM quantification and Fig. 7e). Further analysis of the existing movies is necessary to support this hypothesis. For example, the authors could track *Pcm1*-positive and *Dld*-positive particles in the movies to assess colocalization over time. Additionally, they could track *Pcm1* targeting to the central spindle and its subsequent departure, a behavior expected for *Dld*-positive endosomes as described by the authors. While acknowledging the challenges of tracking particles in live-cell imaging of tissues, it's noteworthy that the authors already track *Dld* endosomes in the current draft (Fig 2c).

2) It is unclear to me how the centrosomes control the asymmetric delivery of *Dld* endosomes, as suggested by the authors in the discussion. The authors show convincingly that *Pcm1* is enriched around the posterior (mother) centrosome, and that *Pcm1* controls the asymmetric trafficking of *Dld* endosomes, but are the two necessarily linked? Could *pcm1* have, somehow, a centrosome-independent function? If *pcm1* is packed in the endosomes, could it be influencing their behaviour independently of the centrosome (i.e., motor activity?). I am no centrosome expert, but the authors would need to somehow manipulate the centrosomes, independently of *pcm1*, to fully support this claim. I understand that this is very challenging to do in vivo, so alternatively they could revisit the discussion section, specifically regarding the role of centrosomes during asymmetric cell division.

3) No rescue experiments were performed to confirm the specificity of the observed phenotype. Although I appreciate that rescue experiments in vivo are technically challenging, the authors have performed such experiments in the past and should

at least check if pcm1-mRNA co-injection rescues the asymmetric Dld distribution phenotype.

Additional comments:

4) The authors should test the statistical significance of all their results. For example, in figure 1, in addition to the pie charts displaying the percentage of each type of division, the authors should calculate the asymmetry index of Pcm1 as they do in Fig 2e as well as in their previous study (Zhao et al. Sci Adv 2021). A statistical analysis is needed here to make the point that “most RGPs had Pcm1 enriched around the posterior centrosome” (line 89). This is also relevant in figure 8c.

5) The Pcm1-GFP and Dld asymmetry data is strong in metaphase and anaphase. However, more data is needed in figure 1j (currently n=12 cells) to justify such a strong sentence as “By telophase, most Pcm1-GFP and Dld endosomes were enriched in the posterior daughter (line 163)”, although I understand that this is a technically-challenging experiment. This also needs a proper statistical analysis (see point 1), as currently 25% of the data points do not agree with that statement. Also, it is interesting to me that the cells dividing with symmetric Dld also display symmetric pcm1-GFP. Could the authors comment on that? This could be an additional argument for pcm1 being transported in Dld-positive endosomes.

6) The authors claim that Pcm1 forms a complex with Par-3 and dynein, but they do not show this experimentally. At most, they show that they are close enough in the same compartment (ED Fig. 5 colP experiments). Unless they show biochemically that these proteins interact with each other there is currently no proof that they form an actual, biochemical, complex.

7) The Dld-positive endosomes are surely diffraction-limited, making the quantification of their number unreliable (ED Fig 2b) and misleading. Additionally, the numbers will highly depend on the imaging conditions and efficiency of injection. Although the data nicely fits the point the authors are trying to make, I would advise against stating that Pcm1 has a role in facilitating endocytosis (line 177). Moreover, the authors do not further develop this point.

8) The data on cell fate assignment is very interesting and beautifully analysed. Could the authors comment on why so many cells divide symmetrically (i.e., 42% P/P) whilst we see before that 75% of the cells have asymmetric Delta (Fig 1j)? What are the other mechanisms at play? This point could probably be developed during the discussion to make the manuscript more accessible to non-experts. Additionally, the authors should explain all the labels present in ED Fig 3. Unless I missed it, I could not find in the legend what “Tel” and “Di” are labelling in panel c.

9) LR-ExM experiments are technically very challenging, but they provide important data that strengthens the manuscript. However, I believe the authors need to better clarify what are they quantifying in Figure 4. What exactly represents “central zone” and what “surrounding cytoplasm” in this case? Is there any difference in anterior-localised endosomes vs posterior? Taking the example cell in Fig. 4a, would it be possible to see a gradient of high levels of Pcm1-Par3-Dld-positive endosomes from the central zone to low levels outside of it (i.e., travelling from anterior, to midzone, to posterior, one should see a clear difference if the exposed hypothesis is correct). This would highly strengthen the current data, as otherwise I am still not fully convinced that pcm1 travels with Dld-positive endosomes (see point 1).

10) The authors claim that “Dld endosomes in the central zone were enriched with Pcm1 and Rab11a (line 332)”. However, unless I am misunderstanding, the data shows that Pcm1+Rab11a+Dld endosomes are not enriched in the central zone vs the surrounding areas (Fig 5b, d). Would it be possible to co-stain for Rab5 and Rab11? Additionally, ED Fig 6 requires statistical analysis to claim that “Pcm1 is associated with Dld recycling endosomes that are decorated with Par-3 and dynein/dynactin complex in the central zone.”

11) The transcriptomic profiling data is somehow not exploited/commented enough in the current manuscript. How are clathrin-coated vesicles are involved in this process? Are the receptors, trafficking molecules and sorting molecules relevant?

Minor:

-(line 51) “Studies in both Drosophila and zebrafish have shown that Notch asymmetry between daughter cells is laid down in the mother via asymmetric distribution of Notch ligand containing endosomes”. This sentence should be rephrased, as asymmetric distribution of Notch-ligand containing endosomes is only one of the mechanisms assuring Notch asymmetry (i.e., it contributes to Notch asymmetry, but it is not the only pathway).

- It is not clear to me from figure 1b and e if the pcm1 signal in the central zone is used to quantify the asymmetry of pcm1 during anaphase division. I would assume it is not, as it is neither anterior nor posterior.

- The authors generate and test N- and C-terminally tagged Pcm1. They found that N-terminal tagged Pcm1 interferes with development, whilst C-terminal does not. (Line 153). I would appreciate seeing the tests and quantifications here (i.e., % survival and/or deformity as in ED Fig. 1e). The authors likely have this data already.

- Red-green-blue palettes (Fig. 1a-c, Fig. 3, Fig. 4, Fig. 5...) should be changed to magenta-green-blue (or other colourblind-friendly combination).

- I believe Fig 8d and Fig 8e references are inverted in lines 503 and 505.

Reviewer #3

(Remarks to the Author)

In this work, Zhao et al. utilized diverse imaging techniques to show that centrosome protein pericentriolar material 1 (Pcm 1) regulates polarized endosome dynamics and RGP fate. This study is highly novel and impactful, as it reveals a novel mechanism by which Pcm 1 conveys centrosome asymmetry. This reviewer is impressed by how the authors used LR-ExM in this study. However, several key points need to be addressed, particularly regarding the acquisition and processing of the LR-ExM images, as these images were critical in reaching the conclusion.

1. The authors used Alexa 488, Dylight 594, and Atto647N for LR-ExM imaging. However, the excitation and emission spectra of Dylight 594 and Atto67N overlap. How did the authors distinguish between the signals of Dylight 594 and Atto647N? Clear identification of proteins is crucial for measuring colocalization. Please provide the imaging setups (i.e., excitation laser wavelengths used and emission filter specifications).
2. In this work, LR-ExM played a critical role in measuring the colocalization of multiple proteins. Therefore, clear identification of true signals from the background or autofluorescence is essential. Considering that antibody signal intensities decrease after expansion due to the volumetric dilution of antigens and the digestion of tags, signal intensities after expansion are not expected to be as high. Therefore, how the images were processed is very important in determining which signals are true. As noted in the Method section, the denoising algorithm was used, but the details of this process, such as the initial raw image (without any processing) before denoising and the image after denoising, are not provided. Including information about how the images were processed, along with the actual images, would help readers understand this manuscript.
3. A biotin-conjugated secondary antibody was used in this work, and streptavidin was applied after digestion. However, this reviewer could not find a procedure for blocking endogenous biotin. Is the level of endogenous biotin not high enough, even without an endogenous biotin-blocking procedure?
4. In LR-ExM, an anti-mouse secondary antibody was applied after digestion. This reviewer is concerned about the nonspecific binding of the secondary antibodies when applied after digestion.
5. The colocalization study is very important for drawing the conclusion in this work. The images in figures 4–6 show that all the proteins have highly punctate structures. Please explain how the colocalization study was performed. What criteria were used to identify true endosome signals and differentiate them from nonspecific signals? Furthermore, what criteria were used to distinguish inside-endosome signals from outside-endosome signals?

Version 1:

Reviewer comments:

Reviewer #1

(Remarks to the Author)

The revised manuscript by Zhao et al. has made significant improvements in addressing previous concerns. However, there are still areas, as highlighted above, that require further clarification and additional data to strengthen the conclusions drawn. Addressing these points would not only refine the current understanding of PCM1's role during neurogenesis but also align the study more closely with established patterns of developmental biology.

1. Distribution of PCM1 in Radial Glia During Neurogenesis in Interphase: The distribution of PCM1 in radial glia during neurogenesis in interphase was not thoroughly discussed in the initial submission or the revised manuscript. A more detailed exploration of PCM1's localization and function during interphase would provide a more comprehensive understanding of its role throughout the cell cycle, not just at metaphase. Addressing this point would enhance the overall narrative of PCM1's role in fate determination of radial glia progenitor cells.
2. Early Localization of PCM1 with CEP83: The authors have responded to previous comments by showing the localization of PCM1 with CEP83 in human iPSC-derived brain organoids. However, this critical information appears relatively late in the manuscript, which might affect the clarity and flow of the narrative. Integrating this data earlier could help in setting the stage for understanding the functional implications of these interactions right from the beginning of the manuscript.
3. Association of PCM1 with the Mother Centrosome: The manuscript discusses the enrichment of PCM1 around the posterior (mother) centrosome at metaphase, marked by CEP83. However, the clarification that PCM1 and CEP83 are concentrated on distinct centrioles within the mother centrosome raises questions about the specificity and implications of this distribution. No data were shown for the localization of CEP38 in zebrafish, which might be crucial for comparing the conservation of this mechanism across species. We suggest to use mRNA injection for localizing CEP38 in zebrafish, as this could substantiate the findings by providing evolutionary context and additional mechanistic insight.
4. Staining with TBR2 and Stage of Forebrain Development: The staining results showing relatively few scattered intermediate progenitor (IP) cells as marked by TBR2 indicate an early stage of forebrain development. This observation seems to contradict the authors' claim about focusing on the neurogenic period of radial glia progenitors. In mammals, the neurogenic period is characterized by a significant number of asymmetric divisions, unlike earlier expansion phases

dominated by symmetric divisions. The manuscript could benefit from a clearer delineation of the developmental stages being studied and how these relate to the broader claims about PCM1's role. It may also be helpful to refine the argument to better align with observed data or expand the developmental timeline studied to cover later stages where asymmetric divisions become more prevalent.

(Remarks on code availability)

The code does not provide a README file.

The repository only has the Python code.

It would be convenient to include example figures to check the reproducibility of the code.

Reviewer #2

(Remarks to the Author)

After a labour-intensive revision by the authors, the study by Zhao et al. convincingly shows that Pcm1 coordinates centrosome and endosome asymmetry during asymmetric cell division. The authors have addressed and satisfactorily answered all of my previous concerns—only two minor issues remain, listed below. I now fully support publication of the revised manuscript in Nature Communications if the first point is addressed.

1) I just realised that in Figure 6c, the authors normalise the intensity of DId fluorescence using DAPI. This is not technically correct, as one cannot normalise a signal from one channel using another channel, especially since DAPI staining and DId immunofluorescence are fundamentally different techniques. Normalisation against Par-3/Dlic1 is also not ideal. One alternative could be to normalise DId intensity against the background signal, but since the movies have been denoised, there is little background left. Another possibility would be to quantify, per endosome, the percentage of surface area occupied by the DId signal, which would be independent of channel and laser power settings. Otherwise, I am afraid the quantification showed in Fig 6c may not yield reliable results due to these limitations.

Additionally, the microscopy section in the Methods is somewhat lacking in detail (e.g., the detector used, the type of immersion oil, how the movies were motion-corrected...). There is also a minor typo: "widefield" instead of "whitefield" in page 24, line 41.

2) The authors are careful in their wording regarding Pcm1 travelling with DId-positive endosomes (e.g., "These [Pcm1] puncta were in proximity to DId endosomes and together they moved toward the posterior side," or "Pcm1 is also localized in the central zone near DId endosomes"). I still think this point could be strengthened by including a kymograph or similar analysis showing Pcm1 being targeted to the central spindle and subsequently departing, using the existing movies. This is not essential for me to support publication, but it would enhance the strength of the paper.

(Remarks on code availability)

I checked that the code is accessible, but I have not tried to run it. It does not include a README file for installation and usage.

Reviewer #3

(Remarks to the Author)

My comments have been fully addressed.

(Remarks on code availability)

Version 2:

Reviewer comments:

Reviewer #1

(Remarks to the Author)

In this round of revisions, the authors made significant improvements. My comments have been addressed satisfactorily, and I now support publication of the revised manuscript.

(Remarks on code availability)

I did not re-review the code. However, I did verify that the repository contains the required information as requested in my previous comments. This includes a clear and organized README file with enough instructions for running the application, as well as example figures to check the reproducibility of the code. This code availability is OK now.

REVIEWER COMMENTS

Reviewer #1 (Remarks to the Author):

Key results:

Zhao et al. aim to demonstrate mechanistically how asymmetric segregation of centrosome-associated proteins during mitosis, specifically PCM1, regulates the fate determination of radial glia progenitor cells. Using different cell- and molecular biological techniques, including live imaging in vivo, expansion microscopy, immuno-assays, and RNAseq, they aim to show that PCM1, in conjunction with Par-3 and Dynein, would mediate the dynamics of endosomes containing Notch-ligands, ultimately leading the acquisitions of different fates by the daughter cells after cell division. Indeed, the roles of asymmetric segregation of centrosomes in (neural) stem cell behaviors remain largely unclear, particularly in mammals. Hence, studies that help clarify the contribution of centrosome inheritance for regulating stem cell systems (here neurogenesis) are essential. Even though the work by the authors has the potential to impact the field significantly, and the data (particularly live imaging) strongly enough support their claims, I recommend addressing the following points before considering the study for publications:

1. The authors mention in the introduction that “how centrosome asymmetry confers daughter cell phase differences at the mechanistic level remains unclear.” The lab of Wieland Huttner has been one of the few addressing the roles of asymmetric inheritance of centrosome-associated structures (i.e., cilia) and mother centrioles in radial glia during mammalian neurogenesis mechanistically. I am surprised these papers (Paridaen, 2013, Cell; Wilsch-Bräuninger, 2012, Development) were not considered, given the relevance for the current manuscript. The protein compositions of the centrosomes have also been shown to play a pivotal role in regulating self-renewal and differentiation of neural stem cells (Camargo Ortega, 2019, Nature; Camargo Ortega and Götz, TICB, 2021; O’Neill, Science, 2021).

We appreciate the reviewer’s comments. We have cited these publications in our revised introduction of the manuscript. They can be found on page 2, lines 16-21 (reference numbers 14, 15, 16, 18, 19).

Paridaen, 2013, Cell; PMID: 24120134; Wilsch-Bräuninger, 2012, Development; PMID: 22096071; Camargo Ortega, 2019, Nature, PMID: 30787442; Camargo Ortega and Götz, TICB, 2021, PMID: 35750615; O’Neill, Science, 2021, PMID: 35709258.

2. The authors comment that the PCM1 is likely enriched around the mother centrioles in metaphase, as the cloud-like distribution around gamma-Tubulin was asymmetric, and this would relate to the role of PCM in cilogenesis. It would be valuable if the authors could indeed demonstrate that PCM1 is enriched in the older centrioles in radial glial

cells, e.g., by co-immunostaining with a distal appendage marker (e.g., CEP83). Notice that subdistal appendages of the mother centriole are retracted during mitosis; thus, stainings with SDA markers (e.g., Ninein, CEP170) are not appropriate.

We appreciate the reviewer's comments. As no good antibodies for CEP83 were available for zebrafish, we performed the experiments in human iPSC-derived brain organoids. Immunostaining of CEP83, α -Tubulin, and PCM1 in day-25 brain organoids uncovered that when PCM1 showed asymmetric distribution on the centrosomes, it was always enriched on the mother centrosome marked by CEP83 (Page 14, lines 42-45, and Fig. 9).

3. Generally, the number of samples for quantification is too low. In Figure 1, the n-sample is mostly below 50 cells. In Figure 1i, 27 cells from 6 embryos were quantified. This is four cells per embryo, which is relatively low for confident statistical analyses.

We have included more cells in the revised manuscript. For Fig. 1 (immunostaining of PCM1), we now included 151 mitotic RGPs. For Fig. 1i-j (in vivo time-lapse imaging of Pcm1-GFP reporter), we typically perform mRNA injection into one blastomere of 8-16-cell stage embryos to achieve sparse labeling. Only ~4-5 mitotic RGPs can be captured per embryo. This is a technically challenging experiment, and 27 cells from 6 embryos took a considerable amount of time and effort. I hope the reviewer agrees that 27 cells from 6 independent experiments are sufficient. We have now included all 27 cells in Fig. 1j for statistical analysis (previously, we only included 12 cells for which the entire mitosis was captured during in vivo live imaging).

4. At the end of the first results section, in the last paragraph, the authors claim to uncover "a previously unknown asymmetry of Pcm1 at the centrosomes and its localization in the central zone near Dld endosomes following enrichment in the posterior daughter." The figures (e.g., Figure 1A) and movies show, however, no clear enrichment at centrosomes in mitosis. They show the presence of Pcm1 protein in centrosomes, as expected, but a significant portion (if not most of it) is not at centrosomes but distributed in the cytoplasm in granules. I would suggest rephrasing this statement or clarifying what they mean by enrichment.

We appreciate the reviewer's comment. I guess the word "enrichment" is ambiguous and could be interpreted as what the reviewer did. We meant that PCM1 showed asymmetric distribution at the two centrosomes. To our knowledge, the asymmetric distribution of Pcm1 at one of the two centrosomes (the mother centrosome) in mitotic RGPs was previously unknown. We have clarified the sentence as "Together, these results uncover a previously unknown asymmetric distribution of Pcm1 at one of the two centrosomes, likely the mother centrosome. Moreover, Pcm1 is also associated with Dld

endosomes in the central zone, and post-division becomes preferentially localized in the posterior daughter". (page 5, lines 1-4).

5. In the first paragraph of the second results section, the authors mention that "perpendicular divisions were reduced with a corresponding increase of non-perpendicular divisions..." While this is true for the *pcm1* KO animals, the same is not the case for *pcm1* morpholinos (Extended data Fig. 2a, 2c). I recommend that the authors address these differences in loss-of-function results.

Absolute numbers of RGPs with designated division patterns were shown in the ED Fig. 2a. Reduced numbers of total dividing RGPs in *pcm1*-deficient conditions made the significance difficult to observe. In the revised ED Fig 2, we added a panel (2b) that quantified the percentage of dividing RGPs with specific division patterns, which showed significant differences in both *Pcm1* MO and *Pcm1* KO compared to controls.

6. The authors determined that there is a relative increase in neuronal production in *pcm1*-deficient fish embryos using clonal analysis. This experiment is appropriate for determining the effects of *pcm1* loss-of-function in brain development. To improve the resonance of their discoveries, particularly during mammalian neurogenesis, I strongly recommend analyzing the loss-of-function impact (short and long-term) during murine cortical neurogenesis. This is because, unlike fishes, mammalian brain neurogenesis uses intermediate progenitor cells to amplify the final neuronal output. This is most prominent in gyrencephalic species, where there are several intermediate progenitors. As a *Pcm1*-full knock-out may be lethal, I would suggest employing shRNAs or miRNA to down-regulate *Pcm1* and track the cells of interest. This experiment could be done in cortical organoids, but this system mostly resembles early human neurogenesis.

The reviewer has raised an interesting and important question regarding the role of *Pcm1* in mammalian neurogenesis, which uses intermediate progenitor cells to amplify the final neuronal output. The developing zebrafish forebrain also has neural progenitors, which, like IPCs, divide to produce two neurons (See Dong et al., Neuron 2012, Fig. 1D, PMID: 22500631). However, a structure like mammalian SVZ has not been reported in zebrafish. Since this manuscript is primarily focused on the role of *Pcm1* in early neurogenesis using live imaging, we feel that the proposed mouse work is out of the current scope. We have discussed the reviewer's point (considering IPCs and mammalian neurogenesis) in the Discussion (Page 17, line 14 - Page 18, line 7).

7. The authors suggest that defective Notch signaling may be behind the *Pcm1* loss-of-function-mediated effects in neurogenesis due to altered *Dld* inheritance. I think this point is crucial, and thus, I would recommend analyzing Notch signaling in control vs. *Pcm1*-LOF conditions using Notch-signaling reporters (e.g., HES-GFP strains).

We have performed HCR in situ of *her4.1* (orthologue of the mammalian *hes* genes) in control, *pcm1* MO, and *pcm1* MO injected with full-length *pcm1* mRNA (rescue). We found a significant decrease of *her4.1* in the forebrain of *pcm1* MO, which was partially rescued by *pcm1* mRNA delivery. These data were added as ED Fig. 3 and Page 5, lines 31-34.

8. The authors use forebrain organoids to compare some of their observations in fish to a human model. Judging by the stainings in the Extended Data Figure 8A, particularly the pattern of TBR2, the results indicate that the organoids have not developed subventricular zones yet, i.e., they are in a relatively early neurogenic stage. If the authors would like to properly compare the role of asymmetric centrosome inheritance in the neurogenesis of mammals, I would – again – recommend checking in murine embryos (e.g., at E14), ferrets, macaques, or further differentiated cortical organoids (in this case, please show Pax6 and Tbr2 stainings together).

We appreciate the reviewer's comments. Understanding the role of Pcm1 in intermediate progenitors of mammalian SVZ is an important future direction but it is outside the scope of this manuscript, which focuses on the role of Pcm1 in ventricular radial glia progenitors during early neurogenesis.

In the revised ED Fig.10, we have shown Pax6 and Tbr2 double immuno-stained section of brain organoids at day 25 and day 30 in culture. We have chosen brain organoids at day 25 in culture for most of our studies because almost all ventricular cells are PAX6 positive at that time. When brain organoids are at day 30 in culture, PAX6-positive cells along the ventricles are fewer than at day 25 in culture.

Reviewer #2 (Remarks to the Author):

In the presented study Zhao and colleagues investigate the role of Pcm1 in regulating asymmetric cell division during zebrafish development.

Using a combination of live cell imaging, developmental biology and high-resolution light microscopy of radial glia progenitors (RGPs) the authors show that Pcm1 is asymmetrically distributed during asymmetric cell division. Intriguingly, the authors show that *pcm1* itself is required for the polarized trafficking of Notch ligand-containing endosomes, a critical process in asymmetric cell division across various species. As a consequence, Pcm1 influences cell fate determination post-division. Capitalizing on expansion microscopy, the authors then characterize the contents of the endosomes, suggesting that *pcm1* is needed for the assembly of the machinery present in the compartment. Finally, the authors show that their findings are conserved in humans using human induced pluripotent stem cells. The authors suggest a conserved

mechanism by which Pcm1 coordinates centrosome asymmetry with endosome dynamics to regulate asymmetric cell division and subsequent cell fate.

Overall, the experiments are well-conducted and the results are intriguing. This study provides novel insights into the connection between cytosolic cell fate determinants and the centrosome, a previously underexplored area in asymmetric cell division. While the work is promising, certain conclusions require additional data or analysis. Although I am enthusiastic about the authors' research, I feel they occasionally overinterpret their findings and make excessive claims. Nevertheless, these issues can often be addressed by further analysing existing data already collected by the authors or tempering conclusions. Therefore, I support publication in Nature Communications provided these points are resolved. Here are some remarks highlighting this:

Major points:

1) I am not fully convinced that pcm1 travels with Dld-positive endosomes. The only live-imaging available to me (Fig. 1h and associated movies) shows a rather independent movement of Pcm1 and Dld signals. This is not essential for the manuscript's main claim that Pcm1 regulates polarized endosome trafficking of Dld-positive endosomes. However, the current draft strongly implies that Pcm1 itself is physically located within the endosomes (based on LR-EM quantification and Fig. 7e). Further analysis of the existing movies is necessary to support this hypothesis. For example, the authors could track Pcm1-positive and Dld-positive particles in the movies to assess colocalization over time. Additionally, they could track Pcm1 targeting to the central spindle and its subsequent departure, a behavior expected for Dld-positive endosomes as described by the authors. While acknowledging the challenges of tracking particles in live-cell imaging of tissues, it's noteworthy that the authors already track Dld endosomes in the current draft (Fig 2c).

We appreciate the reviewer's comments. We also thank the reviewer for acknowledging the co-localization of Pcm1 with Dld endosomes based on the LR-ExM analysis, which, in our opinion, presents the strongest evidence for the association of endogenous Pcm1 with Dld endosomes.

The apparent lack of co-localization of Pcm1 and Dld in our multi-color 3D live imaging is due to the way the live imaging was conducted. For multi-channel Z-stack in vivo time-lapse imaging under the CSW high-speed spinning disk confocal microscope (here, three channels were used: red for membrane, green for Pcm1-GFP, and far red for Dld-Atto647), there are considerable challenges. The two options are:

1) The sequential scanning mode, which is the one we used, scans the entire volume (~30 um) for one channel (e.g., Pcm1) before moving on to the next channel (e.g., Dld). Given the 100 ms exposure per z-plane at a z-step of 1 um, the scanning period for

each channel will take at least 3 s, plus resetting of the objective before scanning the next channel. Based on the live Dld endosome moving speed measured in Fig. 2, we have noticed that Dld endosomes are moving dynamically throughout mitosis. The same endosome could travel 1-2 μm between imaging the Pcm1 channel and the Dld channel. Therefore, it is difficult to assess Pcm1 and Dld colocalization using live scanning data.

2) The interleave scanning mode scans all three channels one by one on the same z-plane and then moves to the next z-plane. While this mode appears ideal for observing co-localization, it takes much longer to complete a whole Z-stack due to the control system setting of the microscope platform, resulting in the scanning interval between each time frame being more than one minute (compared to the time interval of ~ 9 s in the sequential scanning mode). Moreover, the fluorescent signals are bleached only after several time frames, making this option not suitable for our time-lapse imaging. We have now clarified this in the revised methods section (Page 25, lines 8-12).

2) It is unclear to me how the centrosomes control the asymmetric delivery of Dld endosomes, as suggested by the authors in the discussion. The authors show convincingly that Pcm1 is enriched around the posterior (mother) centrosome, and that Pcm1 controls the asymmetric trafficking of Dld endosomes, but are the two necessarily linked? Could pcm1 have, somehow, a centrosome-independent function? If pcm1 is packed in the endosomes, could it be influencing their behaviour independently of the centrosome (i.e., motor activity?). I am no centrosome expert, but the authors would need to somehow manipulate the centrosomes, independently of pcm1, to fully support this claim. I understand that this is very challenging to do in vivo, so alternatively they could revisit the discussion section, specifically regarding the role of centrosomes during asymmetric cell division.

We appreciate the reviewer's comments. We agree with the reviewer that it is possible Pcm1 could influence the endosomal behavior independent of its asymmetric presence at the posterior (mother) centrosome, but the intriguing coordination between its enrichment at the posterior centrosome (detectable at metaphase) and the posterior-directed endosomal movement at anaphase suggests that there is a potential link between these two organelles and the underlying processes.

Indeed, endocytosed Delta traffics through the asymmetrically distributed Rab11-positive recycling endosomes, guided by the centrosome in *Drosophila* sensory organ precursor cells (Emery et al Knoblich Cell 2005). The recycling endosomes interact with mother centriole appendages, which are required for recycling endosomes to the plasma membrane (PMID: 27908937).

We, therefore, propose the following model: Pcm1, through its association with the mother centriole distal appendage protein Cep83, becomes enriched at the posterior

(mother) centrosome in metaphase. Its asymmetric deployment from the posterior centrosome to the central spindle and its association with Dld endosomes and Par-3/Dynein complexes further guide endosome trafficking toward the posterior pole. Future systematic biochemical analysis combined with targeted genetic perturbation and dynamic in vivo imaging are needed to test this model.

We have added all the above viewpoints to the Discussion (Page 16, lines 33-47).

3) No rescue experiments were performed to confirm the specificity of the observed phenotype. Although I appreciate that rescue experiments in vivo are technically challenging, the authors have performed such experiments in the past and should at least check if pcm1-mRNA co-injection rescues the asymmetric Dld distribution phenotype.

Pcm1 mRNA rescue is indeed challenging, as pcm1 is a very large gene. Since the pcm1 MO has been previously validated (PMID: 22767577) and the observed morphant phenotype is like the pcm1 CRISPR KO, we did not perform the mRNA rescue in the first submission. In the revised manuscript, we have added pcm1-mRNA rescue results (See Fig. 2, and ED Fig. 3).

Additional comments:

4) The authors should test the statistical significance of all their results. For example, in figure 1, in addition to the pie charts displaying the percentage of each type of division, the authors should calculate the asymmetry index of Pcm1 as they do in Fig 2e as well as in their previous study (Zhao et al. Sci Adv2021). A statistical analysis is needed here to make the point that “most RGPs had Pcm1 enriched around the posterior centrosome” (line 89). This is also relevant in figure 8c.

We have added the statistics of the asymmetry index in Fig.1 (1e, 1f) in place of the pie chart. We have also added the statistics for asymmetry indices of PCM1, PARD3, RAB11A, and RAB5B for Fig. 8 (8e) and ED Fig. 11 (11d) and ED Fig. 12 (12c).

5) The Pcm1-GFP and Dld asymmetry data is strong in metaphase and anaphase. However, more data is needed in figure 1j (currently n=12 cells) to justify such a strong sentence as “By telophase, most Pcm1-GFP and Dld endosomes were enriched in the posterior daughter (line 163)”, although I understand that this is a technically-challenging experiment. This also needs a proper statistical analysis (see point 1), as currently 25% of the data points do not agree with that statement. Also, it is interesting to me that the cells dividing with symmetric Dld also display symmetric pcm1-GFP. Could the authors comment on that? This could be an additional argument for pcm1 being transported in Dld-positive endosomes.

For Fig. 1J, we have included 15 more cells and there are now 27 in total featuring Pcm1-GFP and Dld distribution at telophase. We performed statistical analysis for the Asy Indices of PCM1-GFP and Dld and found a significant difference between Post. Asy, sym, and anterior Asy (see revised Fig. 1J). We also performed a correlation analysis between PCM1-GFP and Dld and found the coefficient of correlation R to be 0.7767, supporting the association of Pcm1 with Dld endosomes. We have included this in the figure legends for Fig. 1.

6) The authors claim that Pcm1 forms a complex with Par-3 and dynein, but they do not show this experimentally. At most, they show that they are close enough in the same compartment (ED Fig. 5 colP experiments). Unless they show biochemically that these proteins interact with each other there is currently no proof that they form an actual, biochemical, complex.

Our wording “forms a complex with Par-3 and dynein” does not imply a direct physical interaction. Based on our data (both in vivo co-IP and in vivo LR-ExM), we believe that these proteins are in large macromolecular complexes, but not necessarily in direct physical contact. To further enhance the clarity, we have changed our wording to “Pcm1 interacted with Par-3, Dlic1, and Dld, Rab11a, and Rab5b, either directly or indirectly” (Page 9, line 6-7) and “interacting with the polarity regulator Par-3 and dynein motor, either directly or indirectly” (Page 1, line 31-32).

7) The Dld-positive endosomes are surely diffraction-limited, making the quantification of their number unreliable (ED Fig 2b) and misleading. Additionally, the numbers will highly depend on the imaging conditions and efficiency of injection. Although the data nicely fits the point the authors are trying to make, I would advise against stating that Pcm1 has a role in facilitating endocytosis (line 177). Moreover, the authors do not further develop this point.

We have changed the previous ED Fig. 2b (now ED Fig. 2d) Y-axis to read as “internalized Dld-Atto647N dot/RGP”. We have removed the sentence regarding the role of PCM1 in facilitating endocytosis.

8) The data on cell fate assignment is very interesting and beautifully analysed. Could the authors comment on why so many cells divide symmetrically (i.e., 42% P/P) whilst we see before that 75% of the cells have asymmetric Delta (Fig 1j)? What are the other mechanisms at play? This point could probably be developed during the discussion to make the manuscript more accessible to non-experts. Additionally, the authors should explain all the labels present in ED Fig 3. Unless I missed it, I could not find in the legend what “Tel” and “Di” are labelling in panel c.

We appreciate the reviewer’s comments. The main reason for such a difference is that P/P, the division that generated two progenitors, can also be asymmetric, as we have

shown previously (Dong et al., Neuron 2012, Fig. 1D, lineage type 1): the two progenitors can have different proliferative potential in the subsequent round of division. We have clarified this in the revised text (Page 8, lines 11-16).

We have explained the abbreviation Tel and Di in the ED Fig. 3 legends.

9) LR-ExM experiments are technically very challenging, but they provide important data that strengthens the manuscript. However, I believe the authors need to better clarify what are they quantifying in Figure 4. What exactly represents “central zone” and what “surrounding cytoplasm” in this case? Is there any difference in anterior-localised endosomes vs posterior? Taking the example cell in Fig. 4a, would it be possible to see a gradient of high levels of Pcm1-Par3-Dld-positive endosomes from the central zone to low levels outside of it (i.e., travelling from anterior, to midzone, to posterior, one should see a clear difference if the exposed hypothesis is correct). This would highly strengthen the current data, as otherwise I am still not fully convinced that pcm1 travels with Dld-positive endosomes (see point 1).

We appreciate the reviewer’s comments. We have provided a diagram to indicate the central zone and have divided the surrounding cytoplasm into the anterior and posterior parts in anaphase RGPs (Fig. 4b). Our quantification showed that there are significantly more Dld endosomes in the central zone than the posterior or the anterior surrounding cytoplasm (Fig. 4c). The composition of Dld endosomes in the anterior surrounding cytoplasm is significantly different from that of the central zone (Fig. 4d). Since Pcm1-Par3-Dld-positive endosomes can be either Rab5+ (early endosomes) or Rab11+ (recycling endosomes) (See Fig. 5 and ED Fig. 8), because of such heterogeneity, it would be difficult to see a gradient with these markers. (Page 9, line 14-20).

10) The authors claim that “Dld endosomes in the central zone were enriched with Pcm1 and Rab11a (line 332)”. However, unless I am misunderstanding, the data shows that Pcm1+Rab11a+Dld endosomes are not enriched in the central zone vs the surrounding areas (Fig 5b, d). Would it be possible to co-stain for Rab5 and Rab11? Additionally, ED Fig 6 requires statistical analysis to claim that “Pcm1 is associated with Dld recycling endosomes that are decorated with Par-3 and dynein/dynactin complex in the central zone.”

We appreciate the reviewer’s comments. We have clarified the sentence to read as “Quantifications uncovered that Pcm1+Rab5b+ Dld endosomes were significantly depleted in the central zone compared to the anterior surrounding cytoplasm (**Fig. 5c**), whereas Pcm1+Rab11a+ Dld endosomes were enriched in the central zone compared to the anterior surrounding cytoplasm (**Fig. 5f**). These results reveal a relative enrichment of Pcm1+ Dld recycling endosomes in the central zone.” (Page 10, line 13 - 21)

We have searched publications that applied anti-Rab5b and anti-Rab11a IHC experiments. We found both anti-Rab5b and Rab11a antibodies that have been mostly used and verified to work in zebrafish are produced in rabbits. Currently it is hard to perform co-stain of Rab5b and Rab11a with zebrafish samples due to lacking reliable antibodies from different species.

ED Fig. 6 (now ED Fig. 7) served to just show more image examples of LR-ExM RGPs, therefore, we did not provide stats. We have removed the sentence “Pcm1 is associated with Dld recycling endosomes that are decorated with Par-3 and dynein/dynactin complex in the central zone.”

11) The transcriptomic profiling data is somehow not exploited/commented enough in the current manuscript. How are clathrin-coated vesicles are involved in this process? Are the receptors, trafficking molecules and sorting molecules relevant?

We have described the roles of these genes in clathrin-coated vesicle formation and endocytosis, vesicle trafficking, and endosomal sorting. Altered expression of these endocytosis and vesicle transport-related genes in pcm1-deficient embryos support the role of Pcm1 in regulating endocytosis and endosome dynamics. However, given the nature of RNA-seq that cannot distinguish between a direct effect of Pcm1 and an indirect (e.g., compensatory) effect on these pathways, we don't feel much more can be said at this point.

Minor:

-(line 51) “Studies in both Drosophila and zebrafish have shown that Notch asymmetry between daughter cells is laid down in the mother via asymmetric distribution of Notch ligand containing endosomes”. This sentence should be rephrased, as asymmetric distribution of Notch-ligand containing endosomes is only one of the mechanisms assuring Notch asymmetry (i.e., it contributes to Notch asymmetry, but it is not the only pathway).

We have rephrased the sentence as “have shown that asymmetric distribution of Notch ligand-containing endosomes in the mother cell contributes to Notch signaling asymmetry between daughter cells”. (page 2, Line 7-8)

- It is not clear to me from figure 1b and e if the pcm1 signal in the central zone is used to quantify the asymmetry of pcm1 during anaphase division. I would assume it is not, as it is neither anterior nor posterior.

We did not include the central zone in quantifying the asymmetry of Pcm1 during anaphase. Different from mitotic cells at telophase, the two daughter cells are not separated yet. We only used the posterior and anterior surrounding zones for the

quantification. We have included the description in the methods parts of the revised manuscript (page 28, line 27-31).

- The authors generate and test N- and C-terminally tagged Pcm1. They found that N-terminal tagged Pcm1 interferes with development, whilst C-terminal does not. (Line 153). I would appreciate seeing the tests and quantifications here (i.e., % survival and/or deformity as in ED Fig. 1e). The authors likely have this data already.

We have added the statistics of survival in ED Fig. 1g.

- Red-green-blue palettes (Fig. 1a-c, Fig. 3, Fig. 4, Fig. 5...) should be changed to magenta-green-blue (or other colour blind-friendly combination).

We have changed the color palettes accordingly.

- I believe Fig 8d and Fig 8e references are inverted in lines 503 and 505.

We have corrected it.

Reviewer #3 (Remarks to the Author):

In this work, Zhao et al. utilized diverse imaging techniques to show that centrosome protein pericentriolar material 1 (Pcm 1) regulates polarized endosome dynamics and RGP fate. This study is highly novel and impactful, as it reveals a novel mechanism by which Pcm 1 conveys centrosome asymmetry. This reviewer is impressed by how the authors used LR-ExM in this study. However, several key points need to be addressed, particularly regarding the acquisition and processing of the LR-ExM images, as these images were critical in reaching the conclusion.

1. The authors used Alexa 488, Dylight 594, and Atto647N for LR-ExM imaging. However, the excitation and emission spectra of Dylight 594 and Atto647N overlap. How did the authors distinguish between the signals of Dylight 594 and Atto647N? Clear identification of proteins is crucial for measuring colocalization. Please provide the imaging setups (i.e., excitation laser wavelengths used and emission filter specifications).

We used the CSU-W1 Spinning Disk/High-Speed Widefield for imaging ExM samples. We used a laser channel at 561nm for the excitation of Dylight 594 and a laser channel at 647 nm for the excitation of Atto647N. The emission spectra of all fluorescent markers is determined by the band width of the quadrafilters Chroma zET 405/488/561/635 used in the system.

DyLight 594 has an excitation peak at 594 nm and an emission peak at 618 nm. ATTO 647N has an excitation peak at 644 nm and an emission peak at 669 nm.

The 561-filter bandwidth is 573 nm-622 nm. Although Dylight 594 excitation peak is at 594 nm, it can also be excited at 561 nm. Atto 647N would not be excited nor detected by the 561 filter.

The 635-filter bandwidth is >655 nm, which was used to excite and detect Atto 647N. Dylight 594 excitation/emission above 630 nm is almost 0 (less than 2%), which would not cause any mixed emission with Atto 647N.

Dylight 594 has been used and verified in the first paper using LR-ExM. as reported by Dr Shi (J Cell Biol (2021) 220 (9): e202105067.).

2. In this work, LR-ExM played a critical role in measuring the colocalization of multiple proteins. Therefore, clear identification of true signals from the background or autofluorescence is essential. Considering that antibody signal intensities decrease after expansion due to the volumetric dilution of antigens and the digestion of tags, signal intensities after expansion are not expected to be as high. Therefore, how the images were processed is very important in determining which signals are true. As noted in the Method section, the denoising algorithm was used, but the details of this process, such as the initial raw image (without any processing) before denoising and the image after denoising, are not provided. Including information about how the images were processed, along with the actual images, would help readers understand this manuscript.

We appreciate the reviewer's comments. We have applied Aydin, an imaging denoising tool designed for removing unspecific background signals without attenuating true signals.

The Aydin Butterworth denoising algorithm calibrates an optimal Butterworth filter for the given image and its spectrum utilizing self-supervised Noise2Self loss (<http://proceedings.mlr.press/v97/batson19a/batson19a.pdf>). Due to the low-pass nature of Butterworth filters, such an approach successfully removes high-frequency noise components without changing most of the given image spectrum.

For achieving the best denoising effects, the whole raw image dataset with all z-planes from all channels would be required without any modification or crop of raw images. This helps Aydin Butterworth denoiser to calibrate a Butterworth filter for each axis on the entire image. Self-supervised Noise2Self loss works best with raw images given this ensures independence of noise components from actual signal components in the image.

The code and all algorithms can be found on GitHub. Now we have included that information in the revised methods (page 29, lines 14-15). We have also included a

detailed protocol using Aydin for denoising. In the ED Fig. 5a of the revised manuscript, we have shown the raw image of Fig. 4 before denoising.

3. A biotin-conjugated secondary antibody was used in this work, and streptavidin was applied after digestion. However, this reviewer could not find a procedure for blocking endogenous biotin. Is the level of endogenous biotin not high enough, even without an endogenous biotin-blocking procedure?

We have tested endogenous biotin levels in 1 dpf zebrafish embryonic brain sections by using streptavidin only without any primary antibody. All other steps of LR-ExM using biotin-conjugated secondary antibodies were kept the same. The added ED Fig. 5c shows that little background signal of endogenous biotin was detected in LR-ExM brain section samples.

4. In LR-ExM, an anti-mouse secondary antibody was applied after digestion. This reviewer is concerned about the nonspecific binding of the secondary antibodies when applied after digestion.

Thanks for pointing out the error in our previous description. We have added all secondary antibodies before the digestion step. We have corrected it in the part of the LR-ExM protocol in the method of the revised manuscript (page 20, line 29).

5. The colocalization study is very important for drawing the conclusion in this work. The images in figures 4–6 show that all the proteins have highly punctate structures. Please explain how the colocalization study was performed. What criteria were used to identify true endosome signals and differentiate them from nonspecific signals? Furthermore, what criteria were used to distinguish inside-endosome signals from outside-endosome signals?

We measured the colocalization between every two fluorescent channels in the form of Manders' Coefficients (i.e. the proportion of each fluorescence colocalized with the other) by using the JACoP plugin in Image J (Manders et al., 1992). We have cited our protocol paper with detailed steps on how to apply colocalization analyses by using the JACoP plug-in in FIJI in the revised manuscript.

For removing nonspecific signals, the threshold of each channel was set in JACoP using a blank area on the image (i.e., without tissue samples) at first. Then we ran Costes' automatic threshold, which is an algorithm to identify and remove noise using scatter plotting of randomized images generated from the image under analysis. We have provided these explanations in the revised text (Page 25, line 34 to 40).

For LR-ExM images, the raw positive signal intensity would be more than 10 times higher than background noises removed by Aydin on the max-z planes. Based on the

algorithm used in the study, the colocalization coefficients would not be changed significantly after Aydin denoising.

We did not attempt to distinguish inside/outside-endosome signals in this study. Our anti-Dld antibody targets the extracellular domain of the protein, which is expected to be inside the endosome. Both Pcm1 and Par-3 are cytoplasmic, which are expected to be outside the endosome. As the thickness of an endosome membrane is generally considered to be around 7-8 nanometers, we don't think that our method is able to distinguish the inside/outside-endosome signals.

Reviewer #1 (Remarks to the Author):

The revised manuscript by Zhao et al. has made significant improvements in addressing previous concerns. However, there are still areas, as highlighted above, that require further clarification and additional data to strengthen the conclusions drawn. Addressing these points would not only refine the current understanding of PCM1's role during neurogenesis but also align the study more closely with established patterns of developmental biology.

1. Distribution of PCM1 in Radial Glia During Neurogenesis in Interphase: The distribution of PCM1 in radial glia during neurogenesis in interphase was not thoroughly discussed in the initial submission or the revised manuscript. A more detailed exploration of PCM1's localization and function during interphase would provide a more comprehensive understanding of its role throughout the cell cycle, not just at metaphase. Addressing this point would enhance the overall narrative of PCM1's role in fate determination of radial glia progenitor cells.

We appreciate the reviewer's comments. While previous studies have shown that PCM1 primarily localizes to centriolar satellites in interphase cells (e.g., PMID: 12571289, 23345402, 36790165), much less is known about PCM1's dynamic distribution in mitotic cells (from prophase, metaphase, anaphase, to telophase), the focus of this study. Nevertheless, we agree that addressing PCM1's interphase distribution would enhance the understanding of its role in radial glia fate determination.

Additional Data are provided in two ED figures: ED Fig. 1 shows PCM1 expression during interphase in the developing zebrafish forebrain, and ED Fig. 14 shows interphase PCM1 expression in human brain organoids. These data, consistent with previous reports, demonstrate that PCM1 localizes to centriolar satellites during interphase across both model systems.

These data revealed two general patterns of Pcm1 distribution in interphase cells:

1. Tight centrosomal association: In some cells, PCM1 shows concentrated localization directly at the centrosome.

2. Pericentrosomal distribution: In other cells, PCM1 displays a more dispersed pattern around the centrosomal region.

The observed patterns likely represent temporal snapshots of the dynamic Pcm1 distribution captured by immunostaining, suggesting that PCM1 undergoes reorganization during interphase, as we have also observed during mitosis (Fig. 1a). We have added these data and description of Pcm1 distribution pattern at interphase of both zebrafish RGP and human NPCs in the revised manuscript (page 4, line 36-38; and page 14-15, line 44, and Line 1 respectively)

2. Early Localization of PCM1 with CEP83: The authors have responded to previous comments by showing the localization of PCM1 with CEP83 in human iPSC-derived brain organoids. However, this critical information appears relatively late in the manuscript, which might affect the clarity and flow of the narrative. Integrating this data earlier could help in setting the stage for understanding the functional implications of these interactions right from the beginning of the manuscript.

We appreciate the reviewer's comments. We agree that the PCM1-CEP83 colocalization data are important and should be presented earlier in the narrative.

We have restructured the manuscript to integrate this critical information earlier:

1). Enhanced Figure 1: We have expanded Figure 1 to include new immunostaining data showing the preferential association of PCM1 with CEP83-GFP at the mother centrosome in zebrafish (from mRNA-injected embryos), along with γ -tubulin staining to mark centrosomes (revised Fig. 1d).

2). Early Results Integration: The description of PCM1-CEP83 colocalization patterns in zebrafish allowed us to present these data at the beginning of the Results section to establish the relationship between these proteins early in the manuscript.

3). Cross-Model Validation: This zebrafish embryo data demonstrates that an evolutionarily conserved relationship exists between PCM1 and CEP83, strengthening the biological relevance of our findings.

Modifications related to this point can be found at: page 1 (line 29-30); page 2 (line 29-30); page 4 (line 7-13, 30-32); page 16, line 40-41, line 45-47).

3. Association of PCM1 with the Mother Centrosome: The manuscript discusses the enrichment of PCM1 around the posterior (mother) centrosome at metaphase, marked by CEP83. However, the clarification that PCM1 and CEP83 are concentrated on distinct centrioles within the mother centrosome raises questions about the specificity and implications of this distribution. No data were shown for the localization of CEP38 in

zebrafish, which might be crucial for comparing the conservation of this mechanism across species. We suggest to use mRNA injection for localizing CEP38 in zebrafish, as this could substantiate the findings by providing evolutionary context and additional mechanistic insight.

As discussed above, we have addressed the evolutionary conservation question by implementing the recommended CEP83 localization studies in zebrafish. Specifically,

We performed EGFP-CEP83 mRNA injection experiments in zebrafish embryos. Due to the lack of available antibodies for zebrafish CEP83 protein, we cloned mammalian EGFP-CEP83 cDNA from pEGFPC1-CEP83 (Addgene, #128874) and subcloned it into the pcs2+ plasmid vector for mRNA synthesis. The purified EGFP-CEP83 mRNA was injected into zebrafish embryos to visualize CEP83 localization.

Using triple immunostaining with mouse anti-EGFP (Invitrogen MA1-952), rabbit anti- γ -tubulin, and chicken anti-PCM1, we revealed that CEP83 marks the posterior mother centrosome, and Pcm1 is preferentially associated with Cep83 at the posterior mother centrosome (revised Fig. 1d, e), demonstrating evolutionary conservation of this asymmetric distribution pattern.

Regarding the spatial relationship between PCM1 and CEP83 at individual centrioles: While our human brain organoid data clearly demonstrate distinct centriolar localizations of PCM1 and CEP83 within the mother centrosome in mitotic progenitors, the smaller size of zebrafish neurons ($\sim 5 \mu\text{m}$ vs $\sim 10 \mu\text{m}$ for human cells) combined with the resolution limits of spinning disk confocal microscopy, prevents us from definitively resolving individual centrioles in zebrafish radial glia progenitors. However, we do observe that PCM1 and CEP83 immunofluorescent signals show non-overlapping distributions at the posterior mother centrosome, consistent with our human cell findings.

As our primary objective centers on characterizing PCM1's role in asymmetric cell division rather than detailed centrosome ultrastructure, we have not pursued super-resolution approaches to resolve individual centriole architecture. We acknowledge this as an important avenue for future investigation to fully understand the molecular organization of these asymmetric division determinants. (page 18, line 13-20).

The evolutionary conservation of PCM1 asymmetric distribution from zebrafish to humans supports the fundamental importance of this mechanism in vertebrate neural development.

4. Staining with TBR2 and Stage of Forebrain Development: The staining results showing relatively few scattered intermediate progenitor (IP) cells as marked by TBR2

indicate an early stage of forebrain development. This observation seems to contradict the authors' claim about focusing on the neurogenic period of radial glia progenitors. In mammals, the neurogenic period is characterized by a significant number of asymmetric divisions, unlike earlier expansion phases dominated by symmetric divisions. The manuscript could benefit from a clearer delineation of the developmental stages being studied and how these relate to the broader claims about PCM1's role. It may also be helpful to refine the argument to better align with observed data or expand the developmental timeline studied to cover later stages where asymmetric divisions become more prevalent.

We thank the reviewer for this important observation regarding TBR2+ intermediate progenitor (IP) cell numbers and developmental staging. We appreciate the opportunity to clarify our experimental approach and refine our interpretation. Our experiments were conducted using Day 25 forebrain organoids, which correspond to the active neurogenic period based on the literature. During days 25–30, forebrain organoids are in a dynamic and critical phase of neurogenesis, showing rapid development that includes the emergence of neurons and complex neuroepithelial structures. At this stage, the organoids transition from primarily containing neural progenitor cells (NPCs) to a more diverse and organized population of cells (PMID: 40533563; PMID: 27118425).

During the active neurogenic period, ventricular zone (VZ) radial glia progenitors are actively undergoing asymmetric divisions to both self-renew and to produce either a neuron or an intermediate progenitor (IP) marked by TBR2. Importantly, our primary analysis focused on asymmetric cell divisions of radial glia progenitors along the ventricular surface, rather than quantifying TBR2+ intermediate progenitor populations. These cells are likely comparable to the RGPs we study in the developing zebrafish forebrain.

The reviewer correctly notes the relatively sparse TBR2+ IP cells in our organoid system, which is derived from ACD of ventricular progenitors (see above). At Day 25, TBR2+ cells represent intermediate progenitors that typically reside in the subventricular zone (SVZ) and undergo symmetric divisions. The observed IP cell numbers likely reflect organoid-specific developmental dynamics and regional variability, which is commonly reported in 3D organoid systems compared to in vivo development.

Our key findings center on PCM1's role in regulating asymmetric cell divisions of VZ radial glia progenitors verified by using zebrafish embryo and human brain organoid models, which we quantified through both immunostaining and live imaging analysis of Pcm1 distribution and Dld endosomes in mitotic RGPs. We have also done the live tracking of daughter cell fate in the developing zebrafish forebrain. At Day 25, we

observed ~70% of VZ radial glia divisions were PCM1 asymmetric, confirming this as an appropriate developmental window for studying neurogenic asymmetric divisions.

We have clarified these points in the manuscript to better emphasize our focus on VZ radial glia asymmetric divisions as the primary readout for neurogenic activity. (Page 14, line 17-24; page 18, 29-30).

Reviewer #1 (Remarks on code availability):

The code does not provide a README file.

The repository only has the Python code.

It would be convenient to include example figures to check the reproducibility of the code.

We have added a README file with the instructions for each step, and we have also included example figures to demonstrate the whole process. The new file can be found on the updated GitLab link: <https://gitlab.com/bio4212310/kymopy>

Reviewer #2 (Remarks to the Author):

After a labour-intensive revision by the authors, the study by Zhao et al. convincingly shows that Pcm1 coordinates centrosome and endosome asymmetry during asymmetric cell division. The authors have addressed and satisfactorily answered all of my previous concerns—only two minor issues remain, listed below. I now fully support publication of the revised manuscript in Nature Communications if the first point is addressed.

1) I just realised that in Figure 6c, the authors normalise the intensity of DId fluorescence using DAPI. This is not technically correct, as one cannot normalise a signal from one channel using another channel, especially since DAPI staining and DId immunofluorescence are fundamentally different techniques. Normalisation against Par-3/Dlic1 is also not ideal. One alternative could be to normalise DId intensity against the background signal, but since the movies have been denoised, there is little background left. Another possibility would be to quantify, per endosome, the percentage of surface area occupied by the DId signal, which would be independent of channel and laser power settings. Otherwise, I am afraid the quantification showed in Fig 6c may not yield reliable results due to these limitations.

Additionally, the microscopy section in the Methods is somewhat lacking in detail (e.g., the detector used, the type of immersion oil, how the movies were motion-corrected...).

There is also a minor typo: “widefield” instead of “whitefield” in page 24, line 41.

We thank the reviewer for this important technical point regarding our fluorescence quantification methodology in Figure 6c. The reviewer is correct that normalizing Dld fluorescence intensity against DAPI is not fully appropriate, as these represent different staining methods and fluorescence channels with distinct properties. However, we respectfully maintain our approach based on established precedent in the field [e.g., PMID: 34601553 and PMID: 40440177] and keep the statistical results based on DAPI normalization. Because DAPI staining intensity is proportional to DNA content and provides a consistent internal standard among different cells and imaging experiments. DAPI normalization for fluorescence intensity quantification is a widely accepted and frequently used method in cell biology and immunofluorescence studies.

Following the reviewer's suggestion, we also re-analyzed our data using the percentage of endosome surface area (demarcated by either Par-3 or Dlic1) occupied by Dld signal as the primary quantification metric. This approach provides a channel-independent measurement that is not affected by laser power settings or absolute fluorescence intensities.

We used Fiji to define individual endosome boundaries using both Par-3 and Dlic1 staining and measure the area. After that, we measure Dld-positive area within each endosome. We calculated the percentage of the endosome area occupied by the Dld signal.

We have updated Figure 6c by adding the new statistical results and added it into the corresponding methods section to reflect this improved quantification approach. (page 27, line 20-26)

We have added more details about the microscope used in the study and imaging settings in the methods section in the revised manuscript (page 25, line 24-28). More details of the microscope can be found by the link:

<https://microscopy.ucsf.edu/instruments/csu-w1-spinning-diskhigh-speed-widefield>.

Movies for each RGP used for kymograph analyses were registered spatiotemporally using the method we established in our previous work (Zhao et al. Sci. Adv. 2021). It has been addressed in the revised methods (page 26, line 31-42). And it has been included in the “README” of kymopy.

“The spatial registration was done by adopting the center point between two centrosomes as the center of dividing RGPs. The temporal registration was done by adopting the anaphase with the first appearance of the cleavage furrow to be $T = 0$ min. The first appearance of the cleavage furrow was also verified in a set of

embryos with double labeling of the cell membrane and nucleus. Almost all WT (or control MO-injected) RGPs complete their cytokinesis at $T = 2 \text{ min}$ ".

We have corrected the typo of "whitefield" to be "widefield".

2) The authors are careful in their wording regarding Pcm1 travelling with Dld-positive endosomes (e.g., "These [Pcm1] puncta were in proximity to Dld endosomes and together they moved toward the posterior side," or "Pcm1 is also localized in the central zone near Dld endosomes"). I still think this point could be strengthened by including a kymograph or similar analysis showing Pcm1 being targeted to the central spindle and subsequently departing, using the existing movies. This is not essential for me to support publication, but it would enhance the strength of the paper.

We appreciate the reviewer's suggestion to strengthen our claims regarding PCM1 dynamics with Dld-positive endosomes through kymograph analysis. This would indeed provide more definitive evidence for the co-movement we observe.

While we recognize the value of kymographic analysis, several technical constraints prevent us from generating reliable kymographs from our current live imaging data with Pcm1:

1. Lack of centrosomal reference points in Pcm1 live imaging: The live imaging data lack visible centrosomal markers that would serve as fixed reference points necessary for generating accurate kymographs. Without these spatial landmarks, it is impossible to establish reliable directional vectors required for meaningful kymographic analysis.
2. PCM1 Particle Size Heterogeneity: Unlike Dld endosomes, which have relatively uniform and well-defined morphology, PCM1 puncta exhibit considerable size heterogeneity in our imaging conditions. This variability makes it challenging to establish consistent threshold parameters for automated particle tracking, which is essential for reliable kymograph construction.
3. Signal-to-Noise Considerations: The dynamic nature of PCM1 signals, combined with the imaging conditions required to minimize phototoxicity during long-term live imaging, creates signal-to-noise challenges that would compromise kymograph accuracy.

Although we cannot provide definitive kymographic proof due to the absence of centrosomal reference points necessary for accurate directional analysis, our qualitative observations consistently demonstrate PCM1 puncta in close proximity to Dld endosomes during their coordinated movement toward the posterior cell region. This spatial association, combined with co-localization data from expansion microscopy and in vivo co-immunoprecipitation experiments, strongly suggests a functional relationship

during asymmetric inheritance. While our current evidence does not have the quantitative precision of kymographic analysis, it nevertheless provides compelling support for the functional interaction between PCM1 and asymmetric organelle inheritance during radial glia division.

Reviewer #2 (Remarks on code availability):

I checked that the code is accessible, but I have not tried to run it. It does not include a README file for installation and usage.

We have uploaded a README file with all instructions, such as installation and instruction for applying each step with the example figures.

Reviewer #3 (Remarks to the Author):

My comments have been fully addressed.

Response to referees:

REVIEWERS' COMMENTS

Reviewer #1 (Remarks to the Author):

In this round of revisions, the authors made significant improvements. My comments have been addressed satisfactorily, and I now support publication of the revised manuscript.

Reviewer #1 (Remarks on code availability):

I did not re-review the code. However, I did verify that the repository contains the required information as requested in my previous comments. This includes a clear and organized README file with enough instructions for running the application, as well as example figures to check the reproducibility of the code. This code availability is OK now.

Thank you for the comments. We are glad to have your support of the revised manuscript for publication.